 Journal of Cell Biology

# Cross-linker–mediated regulation of actin network organization controls tissue morphogenesis

Daniel Krueger[1,2], Theresa Quinkler[1], Simon Arnold Mortensen[3], Carsten Sachse[3,4], and Stefano De Renzis[1]

**Contraction of cortical actomyosin networks driven by myosin activation controls cell shape changes and tissue morphogenesis during animal development. In vitro studies suggest that contractility also depends on the geometrical organization of actin filaments. Here we analyze the function of actomyosin network topology in vivo using optogenetic stimulation of myosin-II in *Drosophila* embryos. We show that early during cellularization, hexagonally arrayed actomyosin fibers are resilient to myosin-II activation. Actomyosin fibers then acquire a ring-like conformation and become contractile and sensitive to myosin-II. This transition is controlled by Bottleneck, a *Drosophila* unique protein expressed for only a short time during early cellularization, which we show regulates actin bundling. In addition, it requires two opposing actin cross-linkers, Filamin and Fimbrin. Filamin acts synergistically with Bottleneck to facilitate hexagonal patterning, while Fimbrin controls remodeling of the hexagonal network into contractile rings. Thus, actin cross-linking regulates the spatio-temporal organization of actomyosin contraction in vivo, which is critical for tissue morphogenesis.**

## Introduction

Dynamic regulation of the actin cytoskeleton plays a role in many processes requiring cell shape changes, including cytokinesis, cell motility, and morphogenesis of animal tissues (Yam et al., 2007; Martin et al., 2009; Mayer et al., 2010; Sedzinski et al., 2011; Salbreux et al., 2012; Heisenberg and Bellaïche, 2013; Guglielmi et al., 2015). To generate a localized force and modification of cell shape, cortical actin filaments need to be connected to specific plasma membrane domains and contract in a process that depends on the activity of molecular motors such as myosin-II (Levayer and Lecuit, 2012; Murrell et al., 2015; Agarwal and Zaidel-Bar, 2019). In the active phosphorylated conformation, myosin molecules bind to actin filaments and use energy derived from ATP hydrolysis to cause the slide of filaments against one another, resulting in actin network contraction. In addition to myosin activation, in vitro contractility assays and studies in cell culture have highlighted the importance of actin cross-linkers, which connect filaments together and generate networks with different architecture and contractile properties (Svitkina and Borisy, 1999; Pollard and Wu, 2010; Laporte et al., 2012; Reymann et al., 2012; Blanchoin et al., 2014; Chugh et al., 2017; Koenderink and Paluch, 2018). In reconstituted actin networks, when the concentration of cross-linkers is maintained below a certain threshold, actin filaments are loosely interconnected and do not contract.

At higher cross-linker concentrations, actin filaments acquire contractile properties. Above a critical level, they become locked in a noncontractile state (Bendix et al., 2008; Ennomani et al., 2016; Belmonte et al., 2017). Furthermore, it is not only the concentration but also the biochemical characteristics of each individual cross-linker that determine the spatial organization and contraction of actomyosin networks. Low molecular weight cross-linkers tend to promote assembly of actin bundles, whereas cross-linkers with a higher molecular weight favor formation of meshwork-like dendritic networks, which are usually less contractile (Schmoller et al., 2009). However, in vitro contractility assays provide precise information on how organization and contraction of actin filaments respond to specific regulators in a test tube. As yet, there is limited knowledge of how actomyosin networks are organized and regulated in vivo, when actin filaments interact with the plasma membrane and integrate multiple control inputs to achieve precise spatio-temporal regulation of morphogenetic processes.

The early *Drosophila melanogaster* embryo provides a suitable model system for dissecting the molecular mechanisms controlling the structural organization and temporal regulation of actomyosin contraction in an in vivo context (Schejter and Wieschaus, 1993b; Sullivan and Theurkauf, 1995; Mavrakis et al., 2014; Reversi et al., 2014; Xue and Sokac, 2016). After

[1]Developmental Biology Unit, European Molecular Biology Laboratory, Heidelberg, Germany;   [2]Collaboration for joint PhD degree between European Molecular Biology Laboratory and Heidelberg University, Faculty of Biosciences, Heidelberg, Germany;   [3]Structural and Computational Biology Unit, European Molecular Biology Laboratory, Heidelberg, Germany;   [4]Ernst-Ruska Center for Microscopy and Spectroscopy with Electrons (ER-C-3/Structural Biology), Forschungszentrum Jülich, Jülich, Germany.

Correspondence to Stefano De Renzis: derenzis@embl.de.

**Rockefeller University Press**
J. Cell Biol. 2019 Vol. 218 No. 8   2743–2761



fertilization, the embryo develops as a syncytium until the interphase of cycle 14, when the process of cellularization transforms the embryo into a monolayer of 6,000 columnar epithelial cells (Lecuit and Wieschaus, 2000). This morphogenetic process starts with the invagination of the plasma membrane in between the cortically anchored nuclei, which then grows for ∼30 µm into a plane that is perpendicular to the embryo cortex. The assembly of a network of interconnected hexagonal arrays of actomyosin fibers that form at the leading edge of the invaginating plasma membrane is critical for the completion of cellularization (Schejter and Wieschaus, 1993a). To generate the correct cell shape, contractility is restrained until the plasma membrane reaches the base of the nuclei. At this point, the contractile properties and molecular composition of the actomyosin network changes, the levels of myosin-II increase, and the hexagonal network is converted into individual actomyosin rings, which contract and close off the cells basally (Royou et al., 2004; Thomas and Wieschaus, 2004). These changes in actin organization and contractility may be caused by the increase in myosin-II levels occurring over the course of cellularization and/or by additional mechanisms that impact actin organization, as suggested by the identification of the *bottleneck* (*bnk*) mutant phenotype (Schejter and Wieschaus, 1993a).

In *bnk* mutant embryos, actomyosin fibers do not organize in a hexagonal array, and instead constrict prematurely, trapping nuclei and forming short cells (Schejter and Wieschaus, 1993a; Theurkauf, 1994; Reversi et al., 2014). Although Bottleneck protein (Bnk) function is required for embryonic development, its mechanism of action is unknown. The *bnk* gene is expressed at the onset of cellularization and codes for an ∼33-kD intrinsically disordered protein. Bnk localizes to the actomyosin fiber hexagonal arrays during the slow phase of cellularization (the initial 40 min), before being degraded during the fast phase (the last 20 min), when the actomyosin network reorganizes into individual rings and the rate of membrane invagination increases (Reversi et al., 2014). Thus, understanding the molecular mechanisms of action of Bnk may provide insights into the regulatory principles underlying the spatial organization and temporal regulation of actomyosin network contractility during morphogenesis.

In this study, we use optogenetics to activate myosin-II in a spatio-temporally controlled manner and probe actin network responsiveness to myosin-II stimulation. We show that while contractility can be stimulated during the ring phase, the hexagonal network is resistant to myosin-II activation, suggesting that an increase in myosin-II levels alone cannot explain restructuring of the actomyosin network and activation of contractility during cellularization. Biochemical characterization of Bnk function using an in vitro actin cosedimentation assay and EM revealed that Bnk acts as a potent actin bundling protein. Together these results led us to hypothesize that temporal regulation of actomyosin contraction depends on a developmentally controlled switch in actin cross-linking that causes spatial reorganization of actin filaments and contraction. To test whether the molecular principles underlying this hypothesis can be generalized, we interfered with the function of well-established actin cross-linkers during cellularization. Using nanobody-mediated protein knockdown (KD; Caussinus et al., 2011), we identified Cheerio, the *Drosophila* orthologue of Filamin (Li et al., 1999; Sokol and Cooley, 1999), and Fimbrin as two key regulators of actomyosin network restructuring and contractility. While Cheerio is required for establishing the hexagonal pattern of actin filaments, functioning in a similar manner to Bnk, Fimbrin is required for the reorganization of the hexagonal network into contractile rings, thereby counteracting Bnk function. KD of Cheerio in a *bnk* mutant embryo worsened the *bnk* phenotype, while KD of Fimbrin rescued both the *bnk* phenotype and embryonic development. Collectively, our results are consistent with a model in which a switch in actin cross-linking activity coordinates the spatial regulation of actomyosin network organization and temporal control of contractility during tissue remodeling.

## Results

### Actomyosin network organization is differentially sensitive to myosin-II optogenetic stimulation

During the slow phase of *Drosophila* cellularization, cortical actin filaments form an interconnected hexagonal array and do not contract (schematic overview in Fig. 1, A and B). As the plasma membrane is internalized and reaches the base of the nuclei, actomyosin fibers rearrange into individual rings that close off the cells at the base (Fig. 1 C; Schejter and Wieschaus, 1993a). These changes in contractility occur synchronously with an increase in myosin-II levels (Royou et al., 2004). To test whether up-regulation of myosin-II activity is sufficient to drive contractility when actin filaments are organized in a hexagonal conformation, we employed optogenetic stimulation of Rho1 signaling, which triggers myosin-II phosphorylation and activation (Izquierdo et al., 2018). The catalytic domain of the Rho1 exchange factor RhoGEF2 was tagged with the light-sensitive protein domain of CRY2, which interacts with its binding partner CIB1 upon blue light illumination. The N-terminal segment of CIB1, lacking its nuclear localization motifs (CIB1 N-terminal domain [CIBN]), was anchored to the plasma membrane via a CAAX box. In the dark, RhoGEF2-CRY2 is cytoplasmic, and embryonic development proceeded normally (Fig. 1, D–L, nonred boxed area of each panel). To probe contractile responses during cellularization, a region of photoactivation (red box in Fig. 1, D–L) was designed at the level of the actomyosin network, while the neighboring regions of the embryo were imaged exclusively with a 561-nm laser line to visualize myosin-II, and served as an internal control. The following criteria were used to stage embryos during the different phases of cellularization. The priming and the hexagonal phases, occurring during the slow phase of cellularization, corresponded to an ingression of the invaginating furrows from 4 µm to 7 µm, respectively. During the priming phase, the myosin-II signal appeared diffuse, and was not organized in a hexagonal pattern. The ring phase, which marks the beginning of the fast phase, was defined as the time point when the invaginating furrows had passed the base of the nuclei (>10 µm) and myosin-II had acquired a ring-like conformation. In embryos photoactivated when the actin network had acquired a circular conformation, actomyosin rings constricted

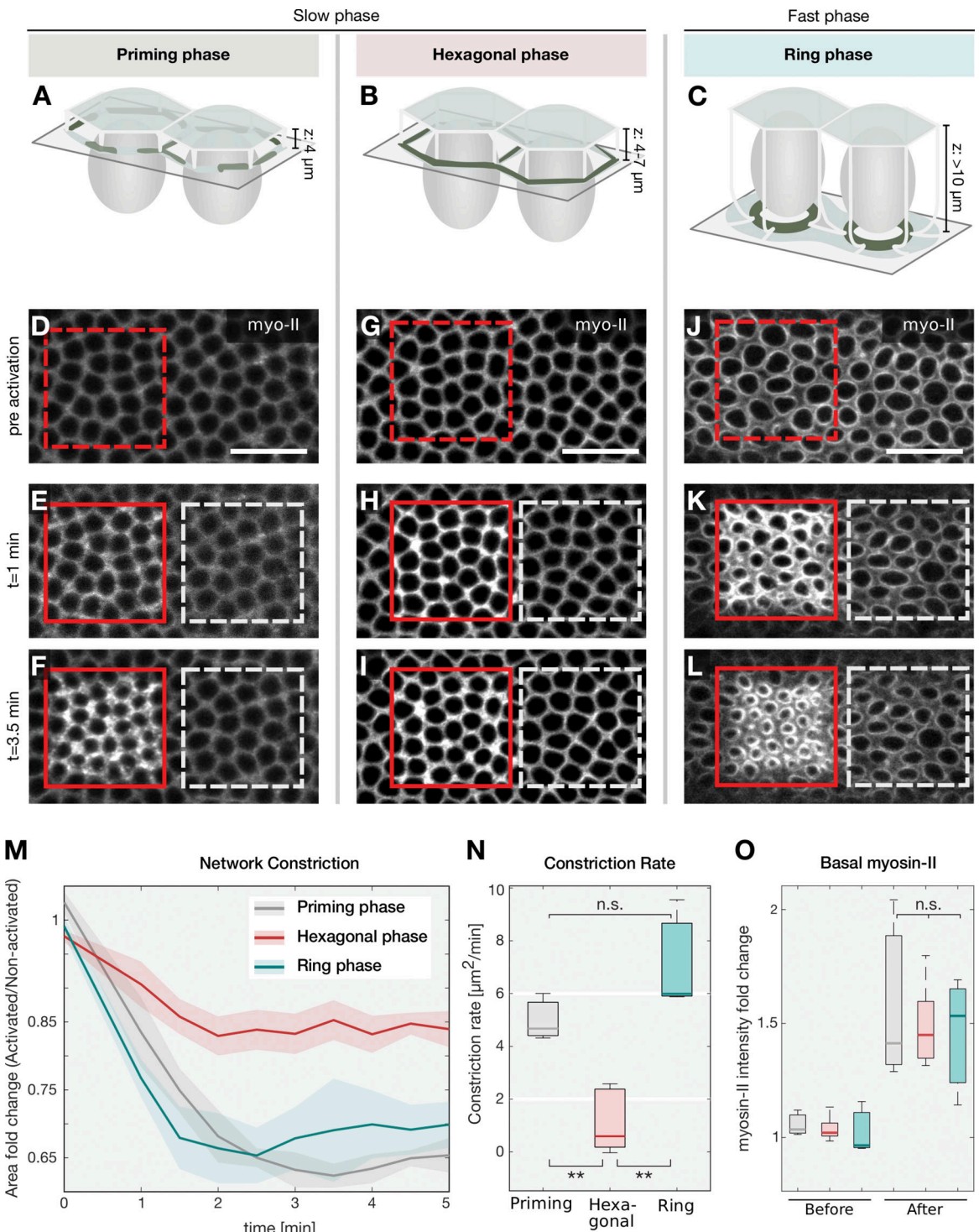

Figure 1. **Actomyosin network configurations are differentially sensitive to myosin-II optogenetic stimulation. (A–C)** Schematic of the experimental setup. *Drosophila* embryos express a RhoGEF2 optogenetic system that stimulates myosin-II activity. A subset of cells is photoactivated specifically at the basal surface (gray rectangle) during different stages of cellularization. During cellularization, the plasma membrane ingresses from the apical surface (semi-transparent light green) toward the interior of the embryo around the nuclei (gray ovals), separating individual cells. The basal actomyosin network (dark green) assembles during the priming phase (ingression depth [z]: 4 µm; A), acquires a hexagonal configuration during the slow phase (ingression depth [z]: 4–7 µm; B), and breaks down into separated contractile rings during the fast phase (ingression depth [z]: >10 µm; C). **(D–L)** Confocal sections of *Drosophila* embryos coexpressing Rho-GEF2-CRY2, CIBN::GFP-pm and the myosin-II probe Sqh::mCh were imaged using a 561-nm laser to visualize the actomyosin network either before photoactivation (D, G, and J), or after photoactivation at the cell base (E, F, H, I, K, and L). Spatially restricted photoactivation was achieved using two-photon excitation (950 nm) in a subset of cells (red square/dashed line: preactivation region; red square/solid line: photoactivated region) during the priming phase (D–F), the hexagonal phase (G–I), or the ring phase (J–L). Upon photoactivation, the myosin-II signal was recorded after 1 min (E, H, and K) and after 3.5 min (F, I, and L). The white dashed squares highlight a region of the same size as of the photoactivation region in the nonactivated part of

the embryo. Scale bars in D–L correspond to 20 µm. **(M)** Quantification of the extent to which the actomyosin network constricted over time upon optogenetic activation of myosin-II during the priming phase (gray, *n* = 3), hexagonal phase (red, *n* = 5), and ring phase (green, *n* = 3). The mean area of individual basal openings within the photoactivated region (number of cells per analyzed embryo: *n* > 20) was measured and normalized to the nonactivated region (number of cells per analyzed embryo: *n* > 30) at the respective time point. While during the priming and the ring phase, the actomyosin network constricted to ~65% of the nonactivated region, during the hexagonal phase it constricted to only 85%. The solid line indicates the mean fold change (area of basal openings in the nonactivated region/activated region), and the semitransparent area represents the SD. **(N)** Network constriction rate calculated during the first minute after photoactivation. When myosin-II was activated during the priming (gray) or the ring (green) phase, the network constricted with a rate >5 µm/min, compared with a constriction rate of ~1 µm/min when myosin-II was activated during the hexagonal phase (red). **(O)** Quantification of myosin-II levels in the photoactivated region before and after photoactivation. Myosin-II levels increased by a factor of ~1.5 upon light stimulation irrespective of network configuration (gray: priming phase; red: hexagonal phase; green: ring phase) as revealed by one-way ANOVA that did not show any significant differences in myosin-II intensity up-regulation in response to photoactivation during the different stages of cellularization ($F[2,8]$ = 0.16, $P$ = 0.8511). **(N and O)** In each box plot, the central mark, the bottom, and the top edge indicate the median and the 25th and 75th percentiles, respectively. Whiskers extend to the most extreme data point. The sample numbers are the same as described in M. For all panels, when indicated, significances were estimated using nonpaired two-sample Student's *t* test with **, $P$ < 0.01; n.s., not significant.

1.5 times more in the photoactivated area than in the nonactivated region (Fig. 1, J–M; and Fig. S1, A and B) with a constriction rate of ~6 µm/min (Fig. 1 N, green box). In contrast, photoactivation during the hexagonal phase resulted in a weaker contractile response that proceeded with a constriction rate of ~1.5 µm/min (Fig. 1, G–I, M, and N). However, if the same embryo was photoactivated during the hexagonal phase and allowed to develop further, a second round of photoactivation caused an approximately twofold increase in actomyosin ring contraction (Fig. S1, C–G). Similar contractile responses could also be induced also if embryos were photoactivated before assembly of the hexagonal network during the priming phase (Fig. 1, D–F, M, and N). These differences in contractility could be caused by the presence of a developmentally controlled RhoGAP activity (Mason et al., 2016), which could prevent myosin-II activation specifically during the hexagonal phase. However, this is unlikely, as the levels of myosin-II were equally up-regulated in response to optogenetic activation during all phases of cellularization (Fig. 1 O). It is also unlikely that myosin-II phosphatase, whose levels were previously shown to be constant until the hexagonal phase and then progressively decrease (Xue and Sokac, 2016), is responsible for these different contractile responses. Optogenetic activation during the priming phase (when myosin phosphatase is present at equal levels as during the hexagonal phase) resulted in a contractile behavior similar to the ring phase. Furthermore, laser ablation of the basal actomyosin network during the hexagonal phase demonstrates no difference in tissue recoil between nonphotoactivated and photoactivated embryos (Fig. 2, A–H; and Video 1), while laser ablation during the ring phase caused a twofold increase in tissue displacement, which increased by ~16% upon optogenetic activation (Fig. 2, I–N; and Video 1). Similarly, initial recoil velocities (i.e., <4 s after laser ablation) did not significantly change during the hexagonal stage, but increased approximately twofold upon optogenetic activation during the ring phase (Fig. 2 O).

Taken together, these results suggest that during the slow phase of cellularization, actin filaments organized in a hexagonal configuration are more resistant to myosin-II–mediated contractile forces than when organized in different patterns during earlier and later stages of cellularization.

## Bnk functions as an actin cross-linker

The unresponsiveness of the hexagonal actin network to myosin-II activation implies that a mechanism must exist to maintain actin filaments in a noncontractile state. We reasoned that Bnk, which is required for hexagonal network organization during the slow phase of cellularization (Schejter and Wieschaus, 1993a), might be involved in the process. Consistent with this hypothesis, optogenetic activation in a *bnk*$^{-/-}$ embryo during early cellularization, when the invaginating furrows have already passed the apices of the nuclei (corresponding to the hexagonal phase in a wild-type embryo), caused the network to constrict to 60% of the initial area, resembling contractile responses during the ring phase of wild-type embryos (Fig. S1, H–M). These results also demonstrate that lack of contractility during the hexagonal phase is not due to mechanical resistance imposed by the nuclei, which deformed in response to optogenetic activation (Fig. 1 K).

Bnk is a hydrophobic protein with large segments of predicted disorder (Fig. S2 A), and due to its insolubility, attempts to produce it as a recombinant protein to study its function in vitro have failed. We took advantage of a previous observation showing that Bnk expressed in HeLa cells (Reversi et al., 2014) localizes to actin-rich structures including stress fibers (Fig. 3, A and B). We expressed a series of deletion constructs in HeLa cells to identify truncated proteins that still localized to stress fibers and presumably retained functionality (Fig. 3, C–H). We also developed an assay based on a custom-made destination vector to identify which of these truncated proteins were soluble when expressed as maltose binding protein (MBP) fusions in *Escherichia coli*. An N-terminal deletion of the first 197 residues of Bnk, Bnk$_{198-303}$, fulfilled both requirements (Fig. 3 F and Fig. S2 B). We purified Bnk$_{198-303}$ as a single soluble protein of the expected molecular weight (Fig. 3 I) and tested if it could bind directly to actin using an actin cosedimentation assay. While Bnk$_{198-303}$ remained in the supernatant upon ultracentrifugation, it shifted to the pellet fraction in the presence of in vitro polymerized actin filaments (Fig. 3 I, blue box), in a similar manner to the actin-binding protein α-actinin (Fig. S2 C).

Given the direct binding of Bnk$_{198-303}$ to actin, we asked whether Bnk could control actin bundling, using a modification of the ultracentrifugation assay. Under low-speed centrifugation, actin filaments did not pellet unless they were bundled together by actin cross-linking proteins. Polymerized actin

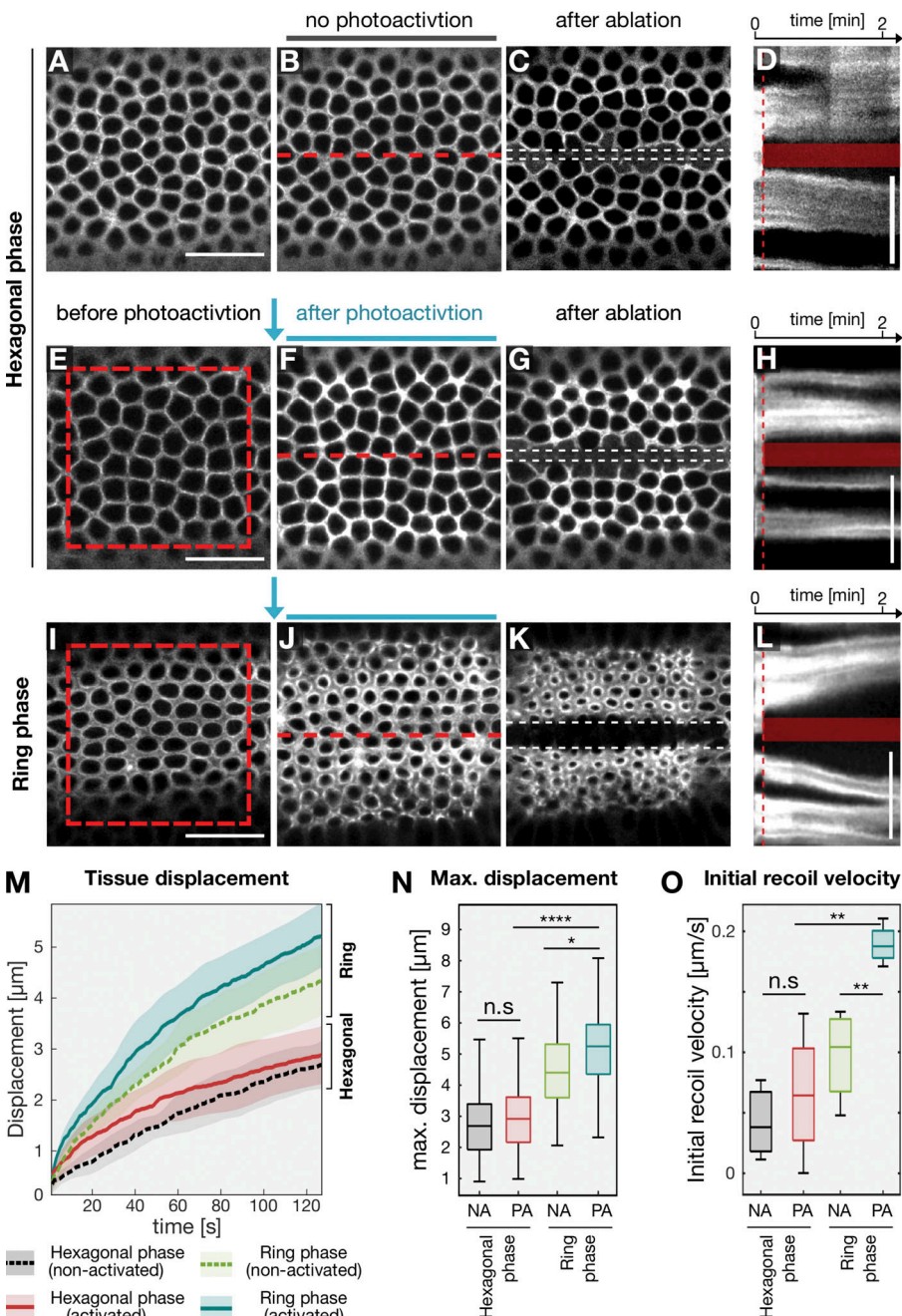

**Figure 2. Actomyosin network configurations display different elastic properties.** Laser ablations were performed in *Drosophila* embryos expressing the RhoGEF2 optogenetic module during the hexagonal phase (A–H) and ring phase (I–L) either without photoactivation (A–D) or after photoactivation (E–L). Panels show the basal actomyosin network at the initial time point (A, E, and I), after photoactivation (F and J), at the corresponding time point without photoactivation (B) or 80 s after laser ablation of the basal network using 800 nm of light (C, G, and K). Scale bars, 20 μm. The position of ablation is indicated by a red dashed line in B, F, and J, and the edges of the recoiled tissue are highlighted by white dashed lines in C, G, and K. **(D, H, and L)** Kymographs showing tissue displacement over time upon laser ablation. Red dashed lines indicate the time at which laser ablation was performed, and the semitransparent box, the ablated region. Scale bars, 10 μm. **(M)** Quantification of tissue displacement after laser ablation during the hexagonal phase (nonactivated: dashed black line, $n = 3$ embryos; after photoactivation: solid red line, $n = 4$ embryos) or during the ring phase (nonactivated: light green dashed line, $n = 4$ embryos; after photoactivation: red solid line, $n = 4$ embryos). Lines indicate mean displacement and semitransparent regions the corresponding SD. At least 10 interfaces were tracked over time per embryos. **(N)** Boxplot showing maximum tissue displacement (at 125 s after laser ablation). Nonactivated (NA) embryos are compared with photoactivated (PA) embryos during the hexagonal phase and ring phase. Optogenetic stimulation of myosin-II did not result in an increase in tissue recoil during the hexagonal phase, while photoactivation caused a significantly higher ($P < 0.05$) displacement during the ring phase. Number of tracked interfaces used for analysis: $n_{Hex:NA} = 36$, $n_{Hex:PA} = 61$, $n_{Ring:NA} = 42$, and $n_{Hex:PA} = 56$. **(O)** Boxplot showing the initial recoil velocity after laser ablation in NA embryos compared with PA embryos during the hexagonal phase and ring phase. The same number of embryos was subjected to analysis as described in M. **(N and O)** In each box plot, the central mark, the bottom, and the top edge indicate the median and the 25th and 75th percentiles, respectively. Whiskers extend to the most extreme data point. For all panels, when indicated, significances were estimated using nonpaired two-sided Student's *t* test with *, $P < 0.05$; **, $P < 0.01$; ****, $P < 0.0001$; and n.s, not significant.

filaments were incubated with Bnk$_{198-303}$ or with BSA as a control before being subjected to low-speed centrifugation. In the presence of Bnk$_{198-303}$, an increased abundance of actin was found in the pellet fraction, consistent with Bnk acting as an actin bundling/cross-linking protein (Fig. 3 J, blue box; and Fig. S2 D). Next, we visualized actin filaments in the presence or absence of Bnk$_{198-303}$ using negative staining EM. Filamentous actin was incubated with or without Bnk$_{198-303}$ and 5 nm Ni-NTA-Nanogold to directly visualize Bnk$_{198-303}$ via a 6xHis tag located at its N terminus (Fig. 3, K–P). In the absence of Bnk$_{198-303}$, actin appeared as single filaments with an average width of ~10 ± 1.6 nm (Fig. 3, K, M, O, and Q). The presence of Bnk$_{198-303}$ induced the formation of thicker actin bundles with an average width of 30 ±10 nm (Fig. 3, L, N, P, and Q), which were decorated by the gold particles (Fig. 3 P, inset). At the concentration used (20 μM), Bnk$_{198-303}$ induced only actin bundling as no other type of actin organization, such as dendritic networks, became apparent. Although we did not test whether different concentrations of Bnk

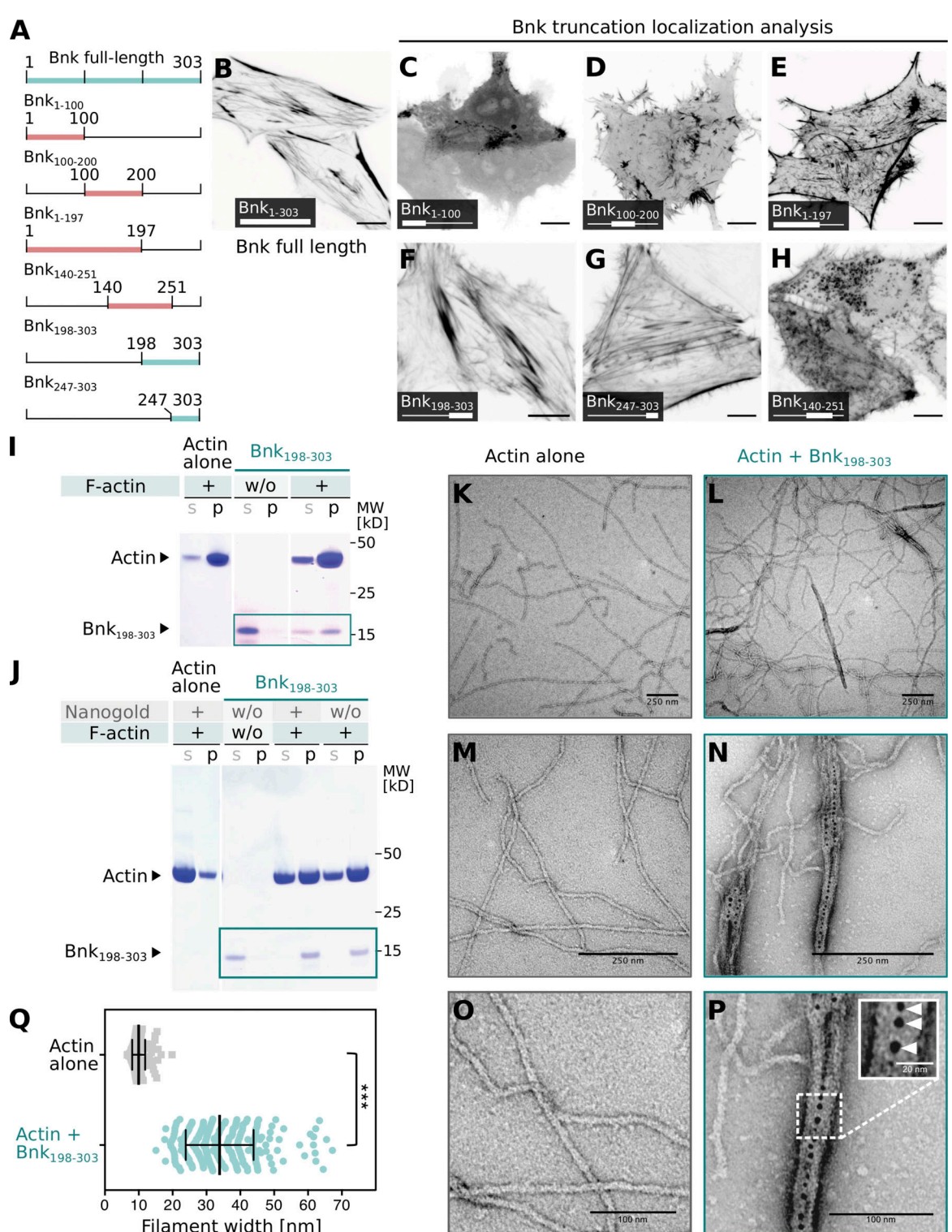

Figure 3. **Bnk protein actin bundling activity localizes to its C-terminal domain. (A–H)** Schematic representation of Bnk truncation constructs expressed in mammalian cells (green: localization similar to full-length Bnk; red: mislocalization; A). GFP-tagged Bnk full-length (B) or truncations of GFP-tagged Bnk with Bnk$_{1-100}$ (C), Bnk$_{100-200}$ (D), Bnk$_{1-197}$ (E), Bnk$_{198-303}$ (F), Bnk$_{247-303}$ (G), and Bnk$_{140-251}$ (H) were expressed in HeLa cells and their localization analyzed using confocal microscopy. Bnk$_{198-303}$ (F) and Bnk$_{247-303}$ (G) closely resembled the localization of the full-length protein. **(I)** Coomassie gel of actin-binding assays. While in the absence of actin, purified Bnk$_{198-303}$ remained in the supernatant fraction (s), in the presence of F-actin, the majority of Bnk$_{198-303}$ protein copelleted with actin filaments (p). The green box highlights Bnk$_{198-303}$ protein bands. **(J)** Coomassie gel of actin-bundling assays. In the presence of F-actin, Bnk$_{198-303}$ shifted from the supernatant (s) to the pellet fraction (p). The ratio of F-actin in the pellet versus supernatant increased in presence of Bnk$_{198-303}$. The presence of 5 nm Ni-NTA Nanogold particles did not alter actin or Bnk behavior. The green box highlights Bnk$_{198-303}$ protein bands. **(K–Q)** Electron micrographs showing actin fibers alone (K, M, and O) or bound by Bnk$_{198-303}$ (L, N, and P) at low magnification (K and L), high magnification (M and N), and a further zoom (O and P). Bnk$_{198-303}$ samples were visualized with gold particles using negative staining. Inset in P shows a zoom of an actin bundle. Arrowheads highlight the localization of Bnk$_{198-303}$ in the center of the bundle. **(Q)** Quantification of the width of actin filaments in absence (gray, n = 200) or presence (green, n = 279) of Bnk$_{198-303}$. Significances were estimated using nonpaired two-sided Student's $t$ test with ***, P < 0.001.

could induce other types of actin organization, the results presented above demonstrate that purified Bnk$_{198-303}$ induces actin bundling/cross-linking. Together with the premature contraction phenotype characteristic of *bnk$^{-/-}$* mutants (Schejter and Wieschaus, 1993a), these results further suggest that Bnk might antagonize myosin-II–mediated contractile forces during the slow phase of cellularization by inducing actin bundling/cross-linking.

## Actin network regulation by actin cross-linkers Cheerio and Fimbrin

We wondered if Bnk's role in controlling the timing of actomyosin contraction during the slow to fast phase transition in cellularization might indicate that actin cross-linkers have a more general role in spatial organization and temporal modulation of actomyosin contractility during morphogenesis. To test this idea, we screened a collection of fly lines homozygous for endogenously YFP-tagged actin cross-linkers (Lye et al., 2014), examining their expression and localization during early embryonic development. We identified two cross-linkers, Cheerio (the *Drosophila* orthologue of human Filamin; Li et al., 1999; Sokol and Cooley, 1999) and Fimbrin, also known as Plastin (de Arruda et al., 1990; Fig. S3 A and B), both of which colocalized with myosin-II to the basal hexagonal network and to actomyosin rings when the network changed conformation (Fig. 4, A–H). Cheerio levels at the base increased over the course of cellularization, reaching a plateau at the end of the hexagonal phase, while Fimbrin levels at the base were stable until the fast phase and decreased toward the end of cellularization (Fig. S3, C–G).

To interfere with Cheerio and Fimbrin function during cellularization, we used deGradFP protein KD (Caussinus et al., 2011), as loss of function alleles are either sterile or not available (Li et al., 1999; Sokol and Cooley, 1999). Anti-GFP nanobody was expressed maternally in females homozygous for either Cheerio::YFP or Fimbrin::YFP (Fig. S3 H). Coexpression of Sqh::mCherry was used to visualize the actomyosin network. Efficient protein KD (<15% of protein control levels) was estimated by confocal microscopy, and only embryos with no detectable YFP signal were analyzed (Fig. S3, I–K). We observed two opposing phenotypes (Fig. 4, I–S; and Videos 2, 3, 4, 5, and 6). Cheerio KD resulted in failure of hexagonal network assembly (Fig. 4, O and P), with premature rounding and constriction (Fig. 4, Q–S; and Videos 3 and 6), resembling *bnk* mutant embryos (Fig. 4, K, L, Q, and R; and Videos 4 and 6). Fimbrin KD, on the other hand, resulted in assembly of a stable hexagonal network that did not change conformation over time (Fig. 4, M, N, Q, and S; and Videos 5 and 6), resulting in a significant delay of actomyosin ring formation (Fig. 4 N) and slower constriction kinetics than controls (Fig. 4 R).

These morphological abnormalities are suggestive of defects in actin bundling/cross-linking. We used nanoscopy to visualize actin organization at higher resolution. Embryos were stained with fluorescently labeled phalloidin and processed for stimulated emission depletion (STED) super-resolution microscopy. In wild-type embryos during the hexagonal phase, actin appeared organized as an array of bundle-like fibers tightly juxtaposed across the entire basal surface of the embryo (although at this resolution [15 nm] we could not unambiguously distinguish between bundled actin filaments or a high concentration of disordered branched networks; Fig. 5, A–C; and Fig. S5, A–C). Over time, actin filaments became organized in individual rings and in a meshwork in the inter-ring spaces (Fig. 5, D–F). In Cheerio- or *bnk*-deficient embryos, actin filaments did not acquire a hexagonal conformation but instead localized in a diffuse meshwork occupying the space separating individual nuclei (Fig. 5, G–L). In contrast, actin filaments in Fimbrin KD embryos did not form rings but remained locked in a stable hexagonal array of actin fibers (Fig. 5, M–R). While during the hexagonal phase the width of actin fibers (measured by segmenting the STED images presented above) in wild-type or in Fimbrin KD embryos was ~200 nm, in *bnk$^{-/-}$* or Cheerio KD embryos, this value dropped to half (Fig. S4, A–E). Segmented actin fibers in wild-type and Fimbrin KD embryos displayed similar actin intensity values, which were significantly higher than in *bnk$^{-/-}$* (approximately twofold) or Cheerio KD embryos (~1.5 fold; Fig. S4 F).

Together these results demonstrate that assembly and restructuring of the basal actin network require the activity of two distinct actin cross-linkers that either facilitate hexagonal patterning (Cheerio) or control remodeling of the hexagonal network into contractile actin rings (Fimbrin).

## The *bnk* mutant phenotype can be rescued by reducing Fimbrin cross-linking activity but worsen by Cheerio depletion

Because of the similarity between the *bnk* and Cheerio KD phenotypes, we tested whether Cheerio is required for Bnk function at the basal actomyosin network and vice versa. We stained Cheerio KD embryos with an anti-Bnk antibody and analyzed the localization of endogenously tagged Cheerio::YFP in *bnk$^{-/-}$* embryos. Although both mutant conditions resulted in a disorganized basal morphology, Bnk and Cheerio correctly localized to the leading edge of the invaginating furrow (Fig. 6, A–F), suggesting that rather than being required for each other's localization, Bnk and Cheerio function in a parallel pathway. Consistent with this interpretation, Cheerio KD *bnk$^{-/-}$* double mutant embryos have a much more severe phenotype than either of the individual mutant phenotypes (Fig. 6, G–Q; Fig. S5 A; and Video 7). In double mutant embryos, actomyosin network constriction initiated at the beginning of cellularization and proceeded with a rate that was approximately twofold higher than in the individual mutants (Fig. 6 J), causing an arrest of membrane invagination apically of the nuclei (Fig. 6, G–Q; and Video 7). Thus, Bnk and Cheerio function synergistically to promote actin bundling and hexagonal patterning.

In contrast, reducing Fimbrin levels in *bnk$^{-/-}$* embryos (Fig. S5, B and C) partially rescued the actomyosin network hexagonal organization (Fig. 7, A–I), with double mutant embryos undergoing normal morphogenesis (Video 8). Both cellularization (Fig. 7 I) and the subsequent ventral furrow invagination during gastrulation occurred normally (Fig. 7, J–L; and Video 9). In ~90% of *bnk$^{-/-}$* embryos, apical myosin-II accumulated in ventral cells, which nonetheless failed to undergo the characteristic cell shape changes that drive ventral furrow invagination (Fig. 7, K and M; and Video 9). The concomitant reduction of Fimbrin

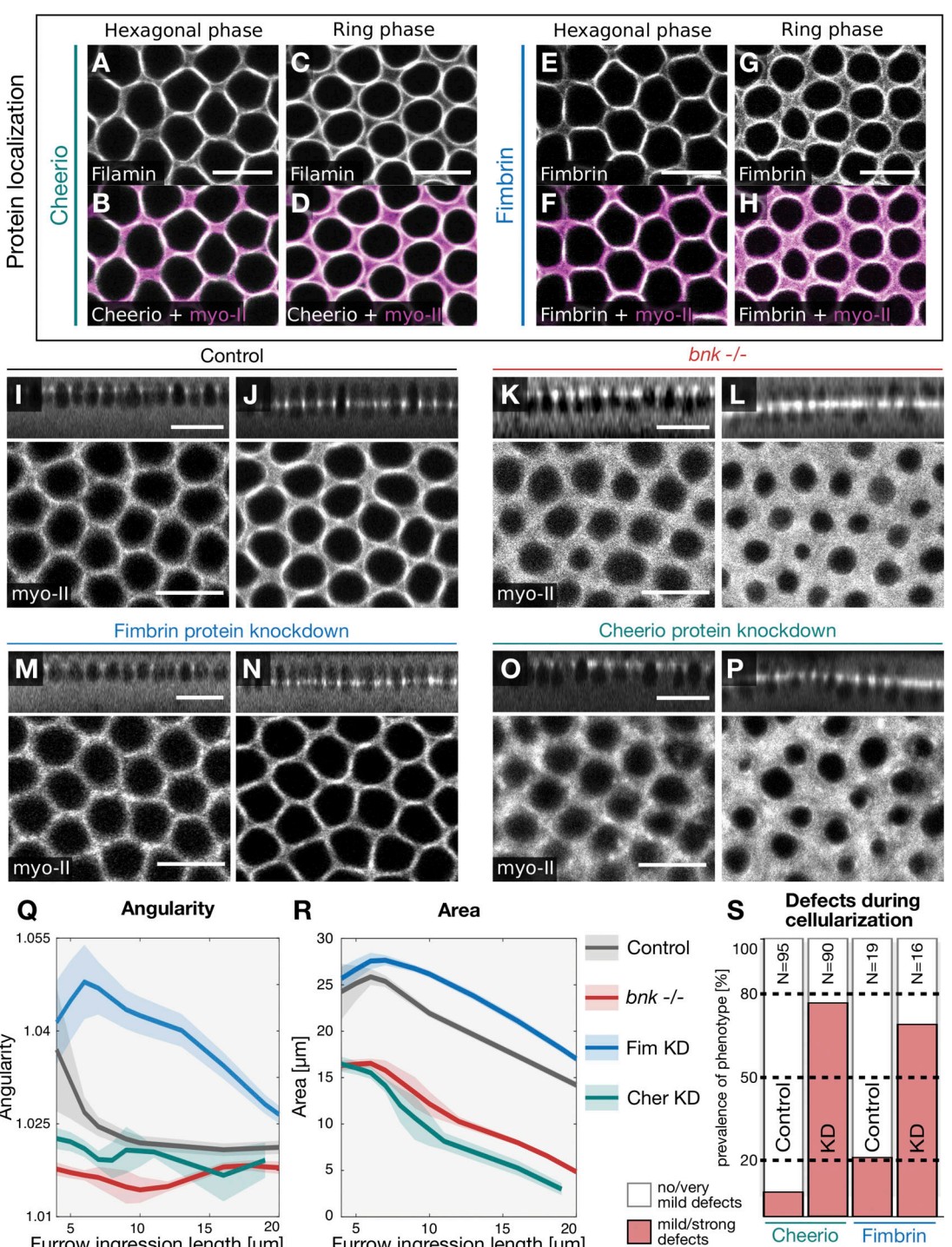

Figure 4. **Actin cross-linkers Cheerio and Fimbrin display distinct regulatory functions during the hexagonal and the ring phase. (A–H)** Confocal sections at the level of the basal actomyosin network during sequential stages of cellularization in embryos coexpressing endogenously tagged Cheerio::YFP (A and C) or Fimbrin::YFP (E and G) and the myosin-II probe Sqh::mCh (magenta; B, D, F, and H). Top view during the hexagonal phase at an ingression depth of 5 µm (A, B, E, and F). Top view during the ring phase at an ingression depth of 12 µm (C, D, G, and H). Both Cheerio and Fimbrin colocalized with myosin-II at the cell base during all stages of cellularization. Scale bars, 10 µm. **(I–P)** Confocal images illustrating the actomyosin network in cellularizing embryos expressing the myosin-II marker Sqh::mCh in control embryos (I and J), in $bnk^{-/-}$ embryos (K and L), and in embryos expressing the nanobody KD module targeting Fimbrin (M and N) or Cheerio (O and P). The upper panels show sagittal cross sections (scale bars, 20 µm) and the lower panels the top view of basal myosin-II signal. Scale bars, 10 µm. Images represent two sequential stages of cellularization: furrow ingression is at 5 µm from the apical surface (I, K, M, and O), and furrow ingression is at 10 µm from the apical surface (J, L, N, and P). In control embryos, the actomyosin network displayed either a hexagonal (I) or round (J) conformation. In $bnk^{-/-}$ or Cheerio KD embryos, the network was disordered and did not acquire the hexagonal configuration (K and O). In contrast, upon Fimbrin KD, the hexagonal phase was prolonged, and the transition into contractile rings was delayed (M). **(Q and R)** Quantification of geometric features of actomyosin network with respect to the ingression depth of the cellularization furrow. The average angularity (Q) and area (R) of cell bases in either a control

(black), *bnk*$^{-/-}$ (red), Cheerio KD (green), or Fimbrin KD (blue) embryo were plotted over the ingression depth. **(Q)** In the control, actomyosin fibers assembled in hexagonal units of high angularity. At an invagination depth of the furrow of ∼7 µm, the hexagonal units rounded up, displaying angularity values closer to 1. In *bnk*$^{-/-}$ and Cheerio KD embryos, actomyosin fibers formed round units (angularity close to 1) from the beginning of the cellularization process. In contrast, the actomyosin network in a Fimbrin KD maintained an angular shape until an ingression depth of ∼15 µm. **(R)** In *bnk*$^{-/-}$ and in Cheerio KD embryos, individual cell bases constricted prematurely at a lower ingression depth compared with control and displayed a smaller area. In the Fimbrin KD condition, network constriction was delayed compared with controls. **(Q and R)** Solid lines indicate average values and semitransparent areas the SEM. For each condition, individual cell openings were segmented and analyzed (basal openings per data point: $27 ≤ n_{control} ≤ 87$, $26 ≤ n_{bnk-/-} ≤ 162$, $15 ≤ n_{CheerioKD} ≤ 42$, $21 ≤ n_{FimbrinKD} ≤ 50$). **(S)** Bar diagram showing the penetrance of Cheerio and Fimbrin KD phenotypes. Cheerio KD ($n = 30$), Fimbrin KD ($n = 16$), and control embryos ($N_{Control\ Cheerio} = 35$, $N_{Control\ Fimbrin} = 19$) were scored for the prevalence of the different phenotypes in blind tests. White color indicates the percentage of embryos with no or very mild phenotypes; red indicates mild or strong phenotypes.

(Fig. S5, B and C) resulted in normal apical constriction, basal nuclear movement, and tissue internalization in 70% of the embryos (Fig. 7, L and M; and Video 9). Taken together, these results support a model in which a switch in antagonizing cross-linking activity regulates changes in actin network organization and contractility during morphogenesis (Fig. 8).

## Discussion

In this study, we have characterized the mechanisms controlling the timing of actomyosin contraction during *Drosophila* tissue morphogenesis. Our results provide evidence that, in addition to myosin-II activation, contractility critically depends on the spatial organization of actin filaments. This spatial organization is developmentally controlled through the activity of actin cross-linkers. By focusing on the dynamic remodeling of a basally localized actomyosin network controlling the process of cellularization at the onset of *Drosophila* embryogenesis, we have identified differential contractile responses to myosin-II stimulation, which correlate with a distinct actomyosin network architecture. In particular, the hexagonal pattern of actomyosin fibers is less responsive to myosin-II stimulation than the meshwork-like conformation that is seen before the hexagonal phase or the ring-shaped conformation seen later in cellularization. This observation is fully consistent with in vitro studies showing that the contractile behavior of actin networks depends on their structural organization (Reymann et al., 2012; Ennomani et al., 2016). This differential sensitivity to myosin-II reflects the need to coordinate contractility and plasma membrane expansion during development in such a way that epithelial cells of the proper shape form and subsequent gastrulation movements can proceed normally.

Based on our results, we propose that the mechanism underlying this regulation involves the modulation of actin cross-linkers. First, we have demonstrated that the key developmental regulator Bnk, which is required for the assembly of the hexagonal arrays, functions as an actin cross-linker promoting actin bundle formation in vitro. Second, we have identified the two actin cross-linkers Cheerio and Fimbrin as being required for actin assembly during the hexagonal phase of cellularization and for actomyosin network remodeling during transition to the ring phase, respectively. While Cheerio and Bnk act synergistically, with double mutants causing severe disorganization of the actin network and an arrest of cellularization, Fimbrin and Bnk display apparently antagonistic functions (see model in Fig. 7).

Cheerio is a relatively large protein known to engage in interactions with multiple proteins and with the plasma membrane (Feng and Walsh, 2004; Zhou et al., 2010). Fimbrin is a small actin cross-linker composed essentially of only two tandem calponin homology domains that confer actin binding (de Arruda et al., 1990). Both Cheerio and Fimbrin are present throughout the process of *Drosophila* cellularization, while Bnk is only transiently expressed during the hexagonal phase (Schejter and Wieschaus, 1993a). Given these data, we suggest that Cheerio controls actin bundling and attachment to the plasma membrane, similarly to its function in ring canal assembly during oogenesis (Li et al., 1999; Sokol and Cooley, 1999). Bnk, which also interacts with plasma membrane phosphoinositides (Reversi et al., 2014), then acts cooperatively with Cheerio to control actin bundling and the interconnections between adjacent hexagonal units. In vitro studies using the vertebrate homologue of Cheerio, Filamin, have demonstrated that it promotes dendritic networks at low concentrations and bundling at higher concentrations (Schmoller et al., 2009). Given the high degree of homology between Cheerio and Filamin (∼50% identity with human; Li et al., 1999; Sokol and Cooley, 1999), one possibility is that the bundling-promoting activity of Cheerio relates to its relatively high concentration during cellularization. The function of Fimbrin may be to cross-link actin filaments in such a way as to allow the transmission of myosin-II forces along actin filaments and breakdown of the interconnected hexagonal arrays into individual contractile rings, once Bnk protein levels decay. In this context, the function of Fimbrin may be similar to its role during *Caenorhabditis elegans* embryogenesis, where it coordinates long-range contractility at the cell cortex (Ding et al., 2017). Our proposed regulatory mode of action for Fimbrin is compatible with a recent study showing that transition from the hexagonal to the ring phase requires myosin-II activity, while actomyosin ring constriction during late cellularization requires actin depolymerization (Xue and Sokac, 2016). Consistent with these observations, our data show that myosin-II stimulation is sufficient to enhance actomyosin ring constriction during only the priming or the ring phase, when Bnk is not present. The Fimbrin mutant phenotype described here resembles the abnormalities of anilin and septin mutant embryos (Thomas and Wieschaus, 2004; Mavrakis et al., 2014; Xue and Sokac, 2016), suggesting that a more complex molecular regulation underlies the spatio-temporal organization of actomyosin network dynamics during cellularization. Our demonstration that the *bnk* mutant phenotype can be rescued by reducing Fimbrin levels suggests that in wild-type embryos Bnk

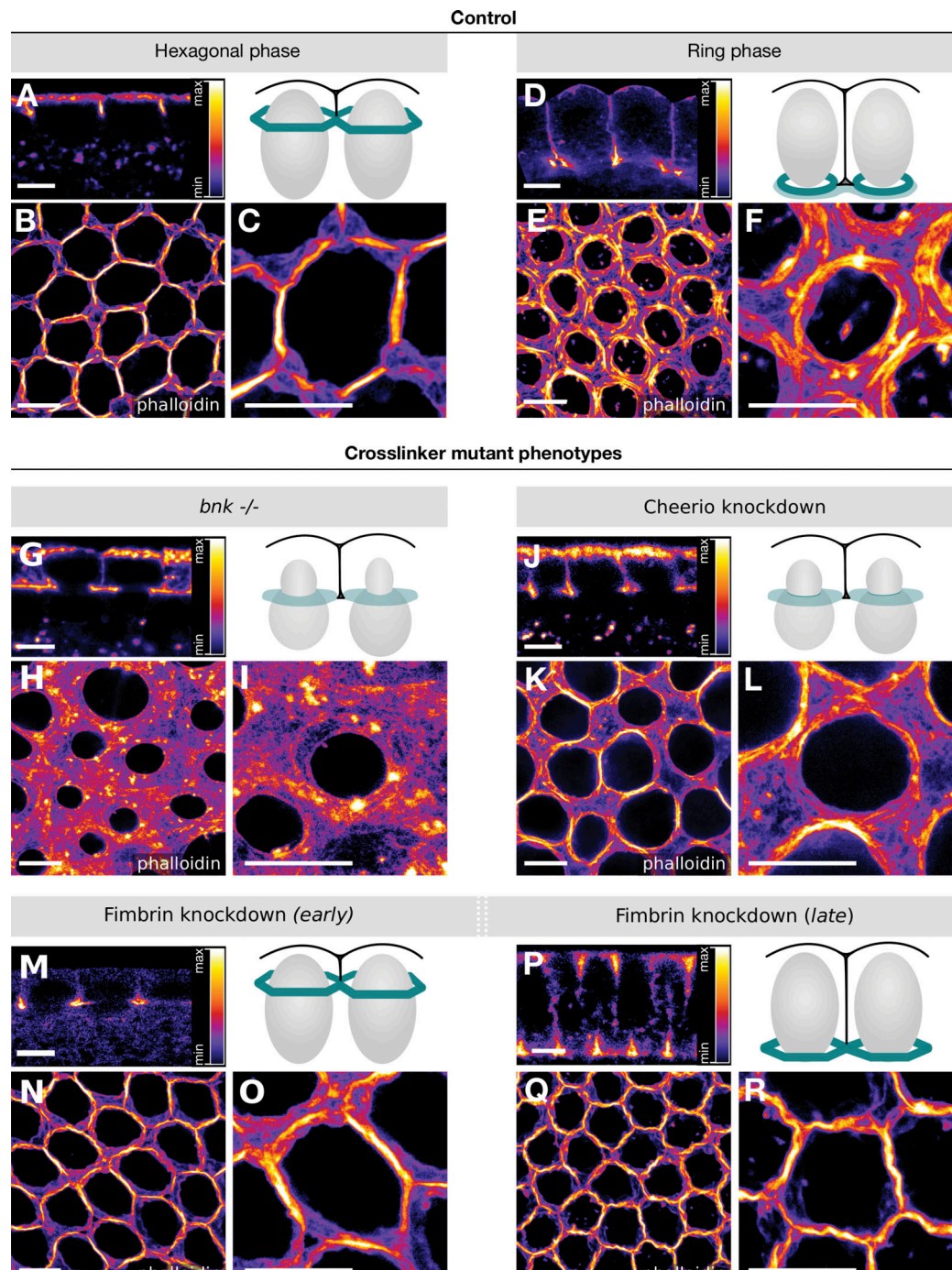

Figure 5. **Super-resolution imaging of Cheerio and *bnk* mutant embryos reveals their critical role in actin bundling during the hexagonal phase. (A–R)** *Drosophila* embryos (wild-type control: A–F; *bnk*−/−: G–I; Cheerio KD: J–L; Fimbrin KD: M–R) were fixed at different stages of cellularization, stained using phalloidin–Atto 647N, and imaged using 2D-STED microscopy at a resolution of ∼15 nm. Grayscale 8-bit still images were pseudo-colored with the *Fire* lookup table (LUT, ImageJ software) to produce false-color images. The sagittal cross sections are shown in A, D, G, J, M, and P, the corresponding basal actin networks in B, E, H, K, N, and Q, and closeups (10 μm × 10 μm) in C, F, I, L, O, and R. The data are shown using the *Fire* lookup table as specified in A, D, G, J, M, and P. Schematics in A, D, G, J, M, and P depict a cross section view of two cells during the corresponding stage of cellularization that was imaged in each respective panel. The basal actomyosin network is colored in green, the nuclei in gray, and the plasma membrane in black. In the wild-type controls (B and C) during the hexagonal phase, actin structures are organized into thick actin fibers. During the ring phase (E and F), actin fibers formed individualized ring structures and a meshwork of actin at the ring interspace. Upon loss of Bnk (G–I), actin fiber formation was severely impaired, and actin was disorganized, forming a meshwork of filaments and thin bundles. KD of Cheerio (J–L) resulted in reduced actin fiber formation, network disorganization, and desynchronized ring constriction. KD of Fimbrin did not affect actin fiber formation during the hexagonal phase (N and O), but rather caused persistence of the actin fiber organized in a hexagonal pattern at later stages of cellularization (Q and R). Scale bars, 5 μm.

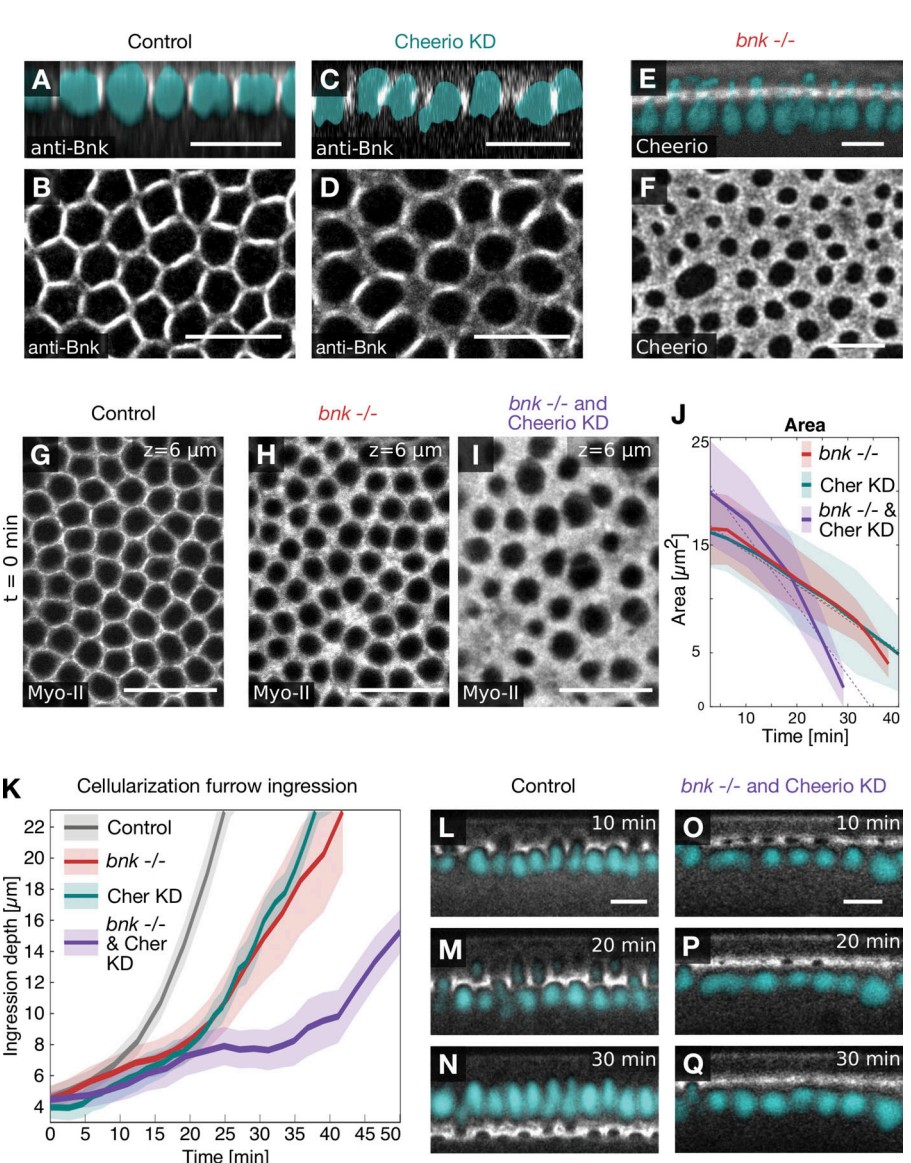

**Figure 6. Bnk and Cheerio act synergistically during hexagonal patterning. (A–D)** Confocal images of Cheerio KD embryos fixed at different stages of cellularization and immunostained using an antibody against Bnk. Sagittal sections showing the localization of Bnk (grayscale) at the leading edge of the cellularization furrow and position of the nuclei (DAPI; cyan) in control (A) and upon Cheerio KD (C). Top view showing the localization of Bnk in control (B) and in Cheerio KD embryos (D), which lack hexagonal patterning. **(E and F)** Localization of Cheerio::YFP in a $bnk^{-/-}$ embryo coexpressing Histone::RPF (cyan) shown in a sagittal section (E) or top view of the basal network (F). Upon loss of Bnk, Cheerio still localized to the cellularization front, although in a diffused pattern due to the lack of actin bundles. Scale bars, 10 µm. **(G–I)** Confocal images of *Drosophila* embryos expressing the myosin-II marker Sqh::mCh in controls (G and K–M), in $bnk^{-/-}$ (H), or in $bnk^{-/-}$ and Cheerio KD double mutant embryos (I and N–P). Top views showing the basal actomyosin network at an ingression depth (z) of 6 µm in a control (G), in a $bnk^{-/-}$ (H), and in a $bnk^{-/-}$ and Cheerio KD double mutant embryo (I). The *bnk* mutant phenotype is characterized by the lack of the hexagonal phase, and hypercontractility worsened in embryos in which Cheerio was also knocked down. Scale bars, 20 µm. **(J)** Graph showing the area of basal openings over time in a $bnk^{-/-}$ (red), a Cheerio KD (green), and a $bnk^{-/-}$ and Cheerio KD double mutant (purple) embryos. A linear function was fitted to the data to estimate the constriction speed. While $bnk^{-/-}$ and Cheerio KD embryos revealed a constriction speed of 0.3 µm²/min, the double mutant constricted with an approximately twofold increased rate (0.65 µm²/min). Solid lines indicate an average value and semitransparent areas the SD. For each condition, individual cell openings were segmented and analyzed (basal openings per data point: 26 ≤ $n_{bnk-/-}$ ≤ 87, 15 ≤ $n_{CheerioKD}$ ≤ 42, 29 ≤ $n_{bnk-/-,CherKD}$ ≤ 91). **(K)** Graph showing kinetics of cellularization (ingression depth of the furrow over time) in controls (black), $bnk^{-/-}$ (red), Cheerio KD (green), and in $bnk^{-/-}$ Cheerio KD double mutant embryos (purple). While in control embryos, the furrow ingressed ~22 µm in ~25 min, in $bnk^{-/-}$ or Cheerio KD embryos, furrow ingression was delayed by a factor of ~1.5. In $bnk^{-/-}$ Cheerio KD double mutants, the ingression kinetics slowed down even further, resulting in a final ingression depth of not >10 µm. Solid lines indicate the average ingression depth at any given time point, and the semitransparent area indicates the corresponding SD. $n_{Control}$ = 5, $n_{Bnk-/-}$ = 4, $n_{CherKD}$ = 8, $n_{Bnk-/-\&CherKD}$ = 5 embryos. **(K–P)** Sagittal sections showing myosin-II (grayscale) in a control embryo 10 min (K), 20 min (L), and 30 min (M) after the cellularization furrow reached an ingression depth of 4 µm. Same analysis as in K–M in a $bnk^{-/-}$ Cheerio KD double mutant embryo at 10 min (N), 20 min (O), and 30 min (P). In double mutant embryos, furrow ingression was severely impaired, resulting in short cells without nuclei. Of note, the position of the nuclei was inferred from lack of myosin-II signal and colored in cyan. Scale bars, 10 µm.

counteracts Fimbrin activity, and that Fimbrin is required for constriction of the actomyosin network. Fimbrin depletion rescues the *bnk* mutant phenotype, including the process of ventral furrow invagination during gastrulation. Tissue invagination in *bnk* mutant embryos could be defective for multiple reasons, but one likely scenario is that the increased basal contractility characteristic of the mutant interferes with apical constriction and the subsequent cell shape changes required for tissue invagination. In agreement with this hypothesis, we have recently demonstrated that increasing myosin-II activity at the basal surface of ventral cells inhibits ventral furrow formation (Krueger et al., 2018).

Bnk may have evolved to specifically regulate contractility during the process of cellularization in the *Drosophila* embryo, as no orthologues have been identified in species other than *Drosophila*. However, its mechanism of action reveals that the temporal regulation of contractile processes during tissue morphogenesis can be encoded in the spatial organization of actin filaments, the dynamics of which are organized by the activity of specific actin cross-linkers. Collectively our results suggest that morphogenesis can be explained not only according to a "gene-centric" framework but also in more general modular terms. In this view, the activity of specific genes during morphogenesis (e.g., *bnk*) that have evolved to fulfill

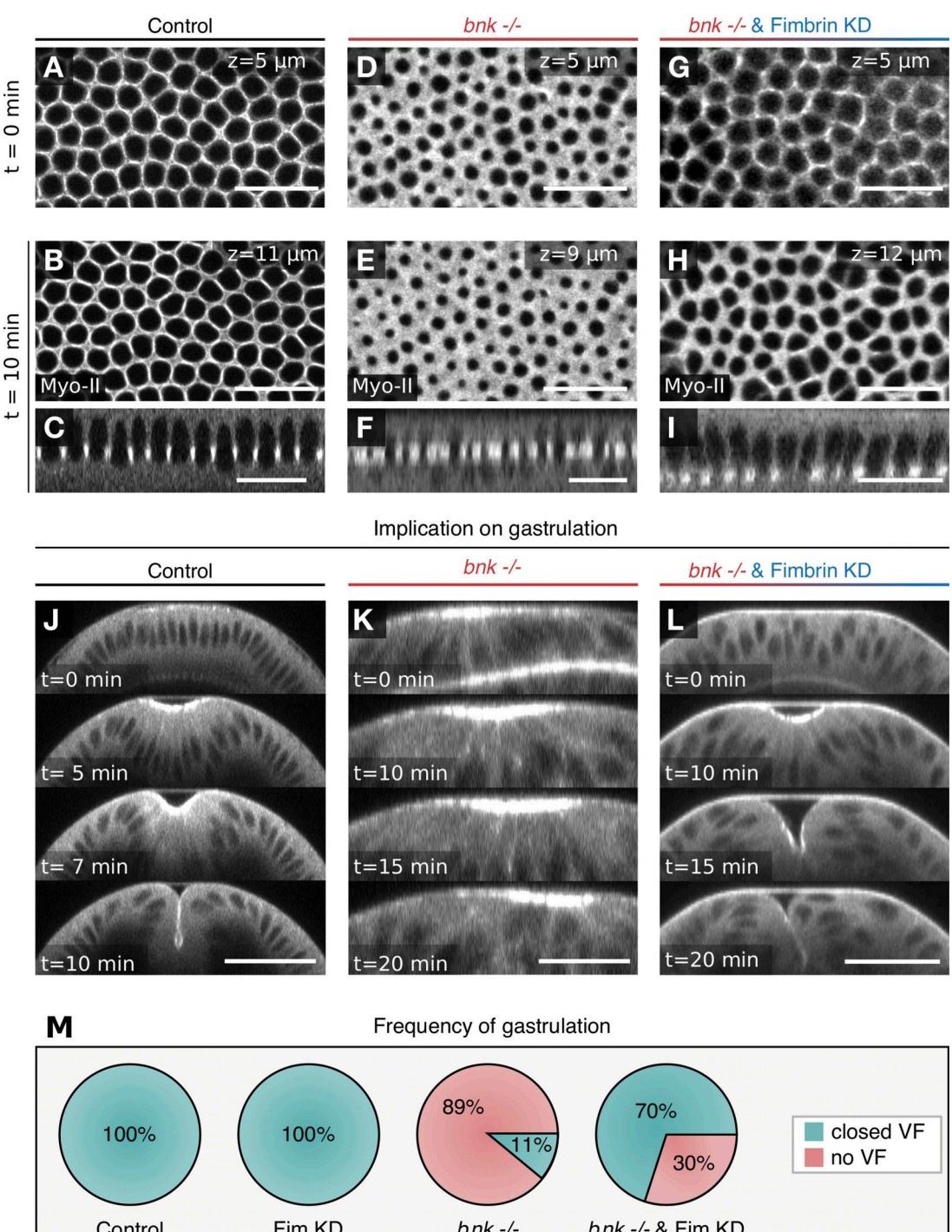

Figure 7. **Depletion of Fimbrin rescues the *bnk* phenotype and gastrulation. (A–I)** Confocal images of *Drosophila* embryos expressing the myosin-II marker Sqh::mCh in control (A–C and J), in *bnk*[−/−] (D–F and K), or in *bnk*[−/−] Fimbrin KD double mutant embryos (G–I and L). Images show top views of the basal actomyosin network at a furrow ingression depth of 5 µm (A, D, and G) and 10 min later (B, E, and H). Images in C, F, and I show sagittal sections of B, E, and H. Depletion of Fimbrin ameliorated the Bnk hypercontractility phenotype and defects in network organization. Scale bars, 20 µm. **(J)** Confocal cross sections of a control embryo expressing the myosin-II marker Sqh::mCherry at the onset of ventral furrow formation, after 5 min, 7 min, and 10 min. **(K)** Confocal cross sections of a *bnk*[−/−] mutant embryo expressing the myosin-II marker Sqh::mCherry at the onset of ventral furrow formation, after 10 min, 15 min, and 20 min. **(L)** Confocal cross sections of a *bnk*[−/−] Fimbrin KD double mutant embryo expressing the myosin-II marker Sqh::mCherry showing rescue of ventral furrow invagination at the onset of ventral furrow formation, after 10 min, 15 min, and 20 min. Scale bars, 10 µm. **(M)** Frequency of normal gastrulation (competed ventral furrow formation) in control (*n* = 19), Fimbrin KD (*n* = 12), *bnk*[−/−] (*n* = 9), and *bnk*[−/−] Fimbrin double mutant (*n* = 10) embryos. Green color indicates the fraction of embryos with normal ventral furrow, and red color indicates the fraction of embryos that initiated but did not complete ventral furrow invagination. Only a small fraction of *bnk* mutant embryos (11%) formed a normal ventral furrow (VF), whereas upon codepletion of Fimbrin, this fraction increased to 70%.

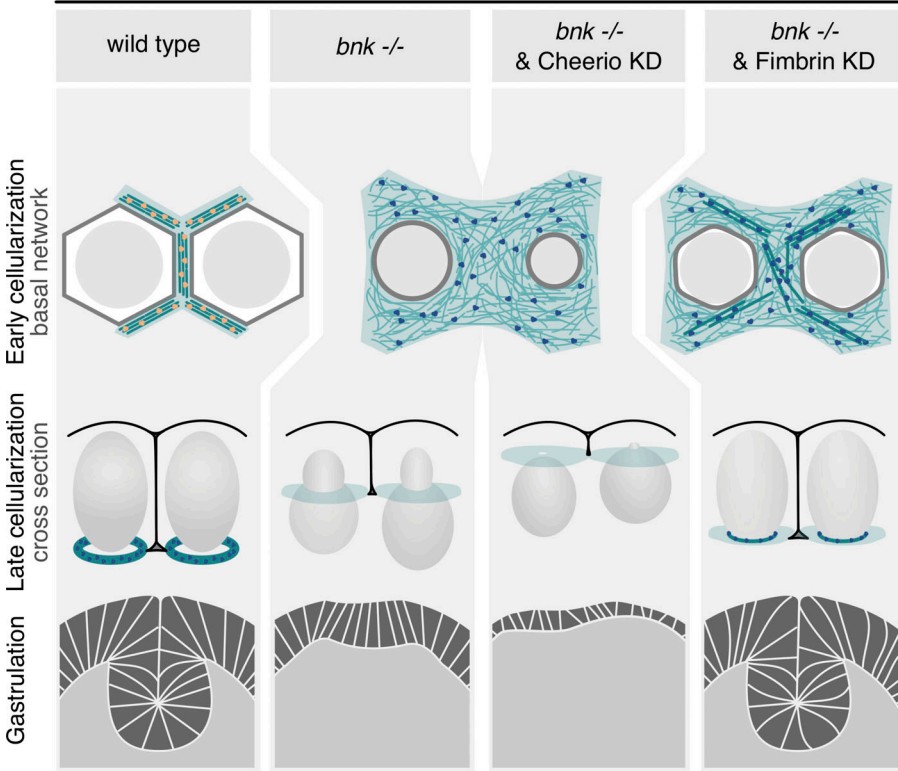

**Synergistic and antagonistic function of actin cross-linkers during morphogenesis**

Figure 8. **Model proposing how the synergistic and antagonistic role of actin cross-linkers control the establishment and remodeling of the basal actomyosin network during *Drosophila* morphogenesis.** In a wild-type embryo during early cellularization, Bnk and Cheerio (yellow dots) organize the basal actomyosin network in noncontractile hexagonal arrays of cross-linked actin filaments (green), which are attached to the plasma membrane (gray hexagon) and surround the nuclei (light gray circles). In addition to cross-linking actin filaments, Bnk and Cheerio may ensure attachment of actin filaments to the plasma membrane. During late cellularization, when Bnk is no longer present, the actomyosin hexagonal arrays break down into individual rings that close off the base of the cells in a process that requires Fimbrin cross-linking activity (blue dots). Fimbrin may function to link actin filaments such that transmission of myosin-II forces transforms the hexagonal arrays in ring-like structures. The correct timing of these events promotes formation of correctly shaped epithelial cells and normal gastrulation movements (ventral furrow invagination). In bnk or Cheerio mutant embryos, lack of actin cross-linking results in a meshwork-like organization of actin filaments and premature constriction. Bnk and Cheerio act synergistically, whereas removal of Fimbrin in bnk$^{-/-}$ embryos rescues the premature contraction phenotype and normal embryonic development can proceed, including gastrulation.

specialized functions can be understood in terms of more generally conserved biochemical processes (e.g., developmental modulation of actin cross-linking).

It will be interesting to explore whether similar mechanisms regulate other contractile processes such as those controlling apical surface dynamics. Apical contractions during tissue morphogenetic events are often pulsatile, with alternating cycles of constriction and stabilization of the plasma membrane (Martin et al., 2009; Solon et al., 2009). Modulation of specific cross-linker activities could be required, for example, to lock the constricted state and avoid relaxation of the apical surface and dissipation of forces.

In conclusion, our results highlight how the functional diversification of actin cross-linkers controls actomyosin network behavior in vivo and provide a striking example of the extent to which the regulated expression of a single cross-linker (Bnk) has an impact on morphogenesis.

## Materials and methods
### Protein expression and purification
To test the solubility of the different Bnk truncations, MBP fusion constructs also containing a His-tag were expressed in *E. coli* Rosetta2 DE3 cells using an auto-induction system. TBL medium (TB and 20% lactose) was inoculated with 5% (vol/vol) overnight culture and incubated for 24–26 h at 22°C. The bacteria culture was harvested by centrifugation and lysed in lysis buffer (20 mM Tris-HCl, pH 7.5, 350 mM NaCl, 0.1% NP-40,

10 µg/ml DNase, 2 mM $MgCl_2$, 20 mM imidazole, 10% glycerol, 2 mM β-mercaptoethanol, and protease inhibitor cocktail; Sigma-Aldrich) using a microfluidizer. Proteins were batch-purified using the Ni-NTA agarose resin (Qiagen) according to the manufacturer's procedure with a 2-h incubation at 4°C, wash buffer 1 (20 mM Tris-HCl, pH 7.5, 350 mM NaCl, 20 mM imidazole, 10% glycerol, and 2 mM β-mercaptoethanol) and elution buffer (20 mM Tris-HCl, pH 7.5, 350 mM NaCl, 200 mM imidazole, 10% glycerol, and 2 mM β-mercaptoethanol). To assess the solubility and aggregation behavior of the purified proteins, a further size exclusion chromatography step was performed using a Superdex 200 Increase 10/300 GL (GE Healthcare) in wash buffer 1. The protein peak at the molecular weight of ∼75 to 50 kD was compared with the void peak.

Recombinant MBP-Bnk$_{198-303}$ was expressed and purified as described above, except purification was done on a cOmplete His-tag purification column (Roche). The MBP tag was cut from MBP-Bnk$_{198-303}$ using a GST-3C protease expressed in-house over night at 4°C, and at the same time the buffer was changed to wash buffer 1, followed by a second cOmplete column His-purification (the His-tag remained attached to Bnk$_{198-303}$). Afterward, a size exclusion chromatography was done using a Superdex 200 16/600 column in the final buffer (20 mM Tris-HCl, pH 7.5, 50 mM NaCl, 10% glycerol, and 2 mM β-mercaptoethanol). The protein was concentrated to 20 µM using Amicon Ultra 4 ml Centrifugal Filters (3,000 nominal molecular weight limit NMWL).

## Actin binding and bundling assay

The actin binding and bundling assays were performed using the Actin Binding Protein Biochem Kit—Non-Muscle Actin (BK013; Cytoskeleton) following the manufacturer's instructions. In brief, the sample protein (Bnk was used at a final concentration of 4 µM) was incubated in the presence or absence of F-actin (16.8 µM actin) for 1 h at room temperature. Afterward, the samples were spun at 100,000 relative centrifugal force (rcf; actin binding assay) or 14,000 rcf (actin bundling assay) for 1 h at 24°C. The supernatant fraction was removed and the pellet resuspended in water. 4× Laemmli buffer was added, and the samples were run on a NuPAGE 4–12% Bis-Tris Gel SDS-PAGE (Thermo Fisher Scientific) and stained with Coomassie brilliant blue.

## Sample preparation for negative stain EM

The samples were prepared according to the described actin bundling protocol, except after 45 min of incubation of F-actin and $Bnk_{198-303}$, the samples were incubated with 5 nm Ni-NTA-Nanogold (Nanoprobes) for 15 min. Excess Nanogold was removed by ultracentrifugation at 49,000 rcf for 45 min at 4°C, and the pellet fraction was resuspended in sample buffer (20 mM Tris-HCl, pH 7.5, 50 mM NaCl, and 10% glycerol). The sample was further diluted (1:5) in sample buffer, and 3.6 µl was applied to a glow-discharged carbon-coated EM grid for 1 min. The sample was blotted away with filter paper, washed with sample buffer and 2% uranyl acetate, and washed once more with sample buffer. Each time a droplet of the respective solution was taken up and the excess liquid was immediately removed with filter paper. A final droplet of 2% uranyl acetate was applied for 2 min before excess of liquid was blotted away, and the grid was dried for 5 min. Grids were imaged using a Morgagni 268 transmission electron microscope (FEI) operated at 100 kV with a side-mounted 1K CCD camera. Filament width was measured using Fiji, and statistical analysis was done using GraphPad Prism 6.0. The filament width for F-actin in the presence of Bnk was only measured when 5 nm Ni-NTA-Nanogold was present in the bundles.

## Live imaging and optogenetics

Flies were kept in a cage with a removable agar plate at the bottom. The agar plate was collected, and halocarbon oil was added to select cellularizing embryos under a standard stereomicroscope. Embryos were dechorionated using 100% sodium hypochlorite for 2 min, rinsed with ddH$_2$O, and mounted onto a 35-mm glass-bottom dish (MatTek) in PBS. For the optogenetics experiments, parental crosses, fly cages, and embryos were kept in the dark, and the sample preparation was done under a red light–emitting LED lamp.

Live-imaging experiments were performed at 20°C using a Zeiss LSM 780 NLO confocal microscope (Carl Zeiss) equipped with a tunable (690–1,040 nm) 140-femtosecond pulsed laser (Chameleon; Coherent) with a repetition rate of 80 MHz and a 40× C-Apochromat (NA 1.20) water immersion objective (Carl Zeiss). A Deep Amber lighting filter (Cabledelight,) was used to filter bright-field illumination to prevent preactivation of photosensitive embryos. Live imaging of YFP-

and GFP-tagged proteins was conducted using 488 nm or 950 nm two-photon excitation, and mCherry-tagged proteins were imaged using 564 nm excitation. The microscope was operated using Zen Black software (Carl Zeiss) and the Pipeline Constructor Macro (Politi et al., 2017 *Preprint*).

For the optogenetics experiments, embryos were photoactivated at different stages of cellularization (priming phase, hexagonal phase, and ring phase). An initial preactivation Sqh::mCherry image stack was acquired using 561 nm excitation. Thereafter, a region of interest was defined within a subset of cells in which RhoGEF2-CRY2 was photoactivated specifically at the cell base using two-photon (950 nm) excitation. The cell base was activated in a focal volume of maximum 2-µm height in a bidirectional scanning mode with a total dwell time of 1 µs, a pixel size of 300 nm, and 10 mW laser power. The duration of photoactivation was adjusted in such a way to cause network constriction during the ring phase (15–50 s). To minimize fluctuation in expression levels of the different optogenetic components, each experiment was completed in 1 d at steady experimental conditions (temperature, etc.). To visualize the effect of photoactivation, the Sqh::mCherry signal was recorded for 5 min after photoactivation. When the same embryo was photoactivated two consecutive times, a new region of activation was set, and the position of the cell base was redefined to activate the cell base using the same parameters as for the first activation. Before and after the second photoactivation, the embryo was imaged using 561 nm laser excitation.

Laser ablations were performed in either control (non-photoactivated) embryos or after optogenetic activation using the same optical setup used for the optogenetic experiments. Laser ablation was achieved by tuning the femtosecond pulsed laser to 800 nm. The region of ablation was set to a line of 500-pixel length (~55 µm) and scanned with a speed of 1.27 µs per pixel and seven iterations. At a time interval of ~1 s, the actomyosin network (Sqh::mCherry) was imaged using a 561-nm laser excitation starting 3 s before the ablation and continuing for 2 min after ablation.

Tissue displacement was analyzed by manually tracking the retracting actomyosin network interfaces lining the ablated region using the Fiji Manual Tracking tool (https://imagej.net/Manual_Tracking). At least 10 individual interfaces were tracked per embryo; for each embryo, the mean tissue displacement was calculated from all tracked interfaces, and at least three embryos were used to calculate the means and SDs of the different experimental conditions tested. The maximum displacement of all tracked interfaces of each experimental condition was measured at the end of the observation period when the displacement curve reached saturation at ~2 min after ablation. The initial recoil velocity was obtained from the slope of a linear function fitted to the displacement over time (1–4 s after laser ablation) of all tracked interfaces of each individual embryo.

## Image analysis and in silico analysis of Bnk protein structure

Images were processed and analyzed using Fiji (https://fiji.sc/) and MATLAB-R2017b (MathWorks). Zeiss LSM files were imported and metadata were extracted using the lsmread function provided via GitHub by Chao-Yuan Yeh (https://github.com/

joe-of-all-trades/lsmread) and the ImageJ Bio-Formats package. The cell base was identified by analyzing the apicobasal intensity profile of the reporter signal using a custom-made MATLAB script. Three to fives slices centered at the peak of the basal signal (using the *findpeak* function) were summed to produce top view representations of, e.g., basal myosin-II.

To measure the ingression depth of the cellularization furrow, the position of the basal myosin-II peak and the cortical fluorescence at the apical surface were used to retrieve cell base and apex, respectively; the distance between them gives the ingression depth. To quantify furrow ingression kinetics for the different mutants, the ingression depth was measured over time and an ingression curve calculated. To account for differences in the start point of image acquisition, the ingression curves were moved along the time axis so that the difference between the curves was minimized to give rise to an average trend line that was zeroed at an ingression depth of 4 µm.

To quantify the area of the basal openings, the basal actomyosin network in the photoactivated and nonactivated region was segmented using a custom-made MATLAB script based on standard Watershed segmentation procedures. Grayscale images were inverted to segment basal openings (appearing as black holes in the positive image). After local contrast enhancement, the gradient magnitude was computed using a Sobel filter. To mark foreground objects (basal openings), a morphological opening (*imopen*) with a disk-shaped structuring element, whose size was iteratively optimized, was applied. Regional maxima were identified and marker edges were cleaned using a closing function followed by an erosion function. The background (black pixel) was computed by thresholding the segmented image and skeletonization. A watershed-based transformation was performed after modifying the gradient magnitude image using the *imimposemin* function with a mask calculated from the segmented fore- and background. A typical segmentation outcome is shown in Fig. S1 A. The area of the segmented basal openings was measured and averaged for the activated and nonactivated regions, respectively, over time and normalized to the initial time point. At least three independent experiments were quantified to give rise to the mean curves presented. The initial constriction rate was calculated by measuring the difference in the area within the first 2 min after initial photoactivation.

To quantify basal myosin-II levels upon photoactivation, the peak of myosin-II signal was identified and the mean intensity of three confocal slices centered at the isolated peak. An average background signal was estimated based on the mean signal intensity inside basal openings and subtracted from the measured value. Fold changes were calculated by normalizing peaks of myosin-II intensity values measured within 4 min after photoactivation to the corresponding values measured before photoactivation.

To measure Cheerio::YFP and Fimbrin::YFP protein levels at the cell base during cellularization, the peak signal at the base was identified, and the mean intensity value of three slices centered around the peak was measured. The average signal intensity of four embryos and the corresponding SD was plotted over the ingression depth of the cellularization furrow. An image stack centered at the basal peak was projected by calculating the mean along the x axis. The so obtained projections were stitched together to produce a kymograph of the cell bases.

To measure the efficiency of the protein KD, the YFP signal of either control embryos or of embryos expressing the anti-GFP nanobody was recorded. The YFP signal intensity profile along the apicobasal cell axis was analyzed to identify the plane with the highest signal. A sum of slices centered at this peak was projected and from that the integrated signal density was measured and normalized to the mean integrated signal density value of control embryos.

To analyze geometric parameters characterizing the actomyosin network in different mutant phenotypes, the basal openings were segmented and the shape of the openings was quantified. The angularity was defined as the square of the quotient of the convex perimeter over the ellipse perimeter (the convex perimeter is the perimeter of the convex hull, and the ellipse perimeter is an ellipse with the same minor and major axis as the analyzed basal opening). The geometric parameters were plotted over ingression depth of the cellularization furrow or time as indicated. For Video 6, the segmented basal openings were laid over the basal myosin-II signal and color-coded according to the respective angularity values for different ingression depths.

The penetrance of Cheerio KD and Fimbrin KD mutant phenotypes was evaluated in a blinded controlled manner by visual assessment of embryos during cellularization scoring for either network distortion in combination with premature constrictions (Cheerio KD phenotype) or prolonged persistence of network angularity (Fimbrin KD phenotype).

In silico analysis based on the primary structure of Bnk were done using RONN v3.2 (Yang et al., 2005) for the per-residue disorder prediction and ProtScale (ExPASy) to generate the Kyte and Doolittle hydrophobicity plot with a window size of 9. The mRNA structure of Cheerio and Fimbrin was analyzed using Ensembl, and the protein domain architecture was analyzed using SMART.

**STED microscopy**

2D-STED microscopy was performed on a Leica SP8 confocal microscope (DMI6000) with a 100× oil HC PL APO CS2 objective (NA 1.40) and type F immersion liquid at 22.5°C using LAS X (Leica) software. The phalloidin-atto647N–stained sample was imaged in both normal confocal mode using only a 633-nm excitation and STED mode combining 633-nm excitation with the 775-nm STED laser. Photons in a range of 640 to 750 nm wavelength were detected using a Leica HyD detector (6% gain). The image was scanned with a pixel size of 15 nm, an averaging of 16, a dwell time of 1.2 µs, and a pinhole of 0.93 A.U. The depicted cross sections were imaged using a reduced resolution.

Actin fibers from STED images were segmented using the open-source MATLAB package Stress Fiber Extractor [SFEX]; Zhang et al., 2017). In the "thick stress fiber" mode, the following parameters were selected: (LFT&OFT) filter radius: 5; number of angles: 20; threshold: 0.4; junction removal: 1; short fragment filter: 10; single iteration; tip search angle, radius: 60, 60; default grouping condition; filter for short ungrouped filaments: 60;

lower bound: 0.3; upper bound: 10 (this value had to be reduced to 5 for $bnk^{-/-}$ embryos to achieve accurate segmentation). The width of segmented fibers was measured by SFEX. To analyze actin density, STED images were background-subtracted (background was inferred from the mean intensity of basal openings), and segmented fibers were used as masks to measure the mean phalloidin intensity. To estimate the fibers' density, actin intensity values of each fiber were divided by the intensity of a manually segmented single fiber. For a relative comparison of fiber densities among all tested conditions, fiber density values were divided by the median value measured during the hexagonal phase in control embryos.

### Statistical analysis
Statistical analyses were performed in MATLAB. Unpaired two-sample Student's $t$ test was performed to determine if two sets of data were significantly different from each other, and the $P$ value was calculated. To compare multiple samples and test for significant differences, a one-way ANOVA was performed. Data distribution was assumed to be normal, but this was not formally tested.

### Anti-GFP nanobody-mediated protein KD
Female flies homozygous for YFP insertion in the locus of *cheerio* or *fimbrin* and expressing the myosin-II probe Sqh::mCh were crossed to males heterozygous for the respective YFP-tagged insertion, and expressing the anti GFP-nanobody under the control of the maternal tubulin promoter, to generate flies homozygous for either YFP-*cheerio* or *fimbrin* and coexpressing Sqh::mCh and the anti-GFP nanobody. Heterozygous flies (of the parental crosses) expressing YFP-tagged *cheerio* or *fimbrin* together with the nanobody were kept at 25°C to reduce the efficiency of protein KD and minimize the selective pressure to compensate for the cross-linker depletion. Flies of the final genotype were collected for 3 d, put in a cage, and shifted to 18°C, and after 1 d of adaptation they were used for phenotypic characterization over a period of 5 d. Flies having a balancer chromosome instead of the anti-GFP nanobody-containing chromosome served as controls.

### Actin staining
Flies were kept in a cage with a removable agar plate supplemented with apple juice and yeast paste at the bottom at 18°C overnight. The agar plate was collected and the yeast paste removed. Halocarbon oil was added and embryos of the desired cellularization stage selected. Embryos were bleached using 100% sodium hypochlorite for 1 min, rinsed three times in ddH$_2$O, and fixed in formaldehyde-saturated heptane for 40 min. The formaldehyde-saturated heptane solution was the upper phase separated from a mixture of equal amounts of 100% heptane with 37% formaldehyde vigorously shaken for 15 min. Fixed embryos were placed onto two-sided sticky tape and covered with PBS. Using forceps and brushes, the vitelline membrane was removed from the embryos under a standard stereomicroscope, and the embryos were collected in PBS and 0.1% Triton X-100 in an Eppendorf tube. Devitellinized embryos were washed three times in PBT (PBS, 1% BSA, and 0.05% Triton

X-100) and incubated in blocking solution (PBS, 6% BSA, and 0.05% Triton X-100) for 1 h at room temperature. Afterward, embryos were incubated in actin staining solution (15 μl of phalloidin-atto647N stock solution [20 μM in methanol, kept at −20°C] in 1 ml of PBS and 0.1% Triton X-100) for 1.5 h at room temperature and washed three times in PBT. Embryos were mounted on a glass slide by removing the PBT solution and adding ProLong Gold Antifade Mountant (Molecular Probes/Thermo Fisher Scientific). A 0.16–0.19-mm-thick (thickness 1.5) cover glass (Glaswarenfabrik Karl Hecht GmbH & Co KG) was placed on top, and excessive mounting medium was removed using tissue paper, dried, and sealed using nail polish.

### Immunostaining
Embryos were collected after incubation of the fly cages at 18°C for 8 h. Fixation of the dechorionated embryos was done in a mixture of heptane and 4% paraformaldehyde/PBS (Electron Microscopy Sciences). Primary antibody against GFP (1:1,000; mouse anti-GFP; Torrey Pines) and against Bnk (1:200; rat anti-Bnk; Reversi et al., 2014) were used. Fluorescently labeled secondary antibody (mouse–Alexa Fluor 488, rat–Alexa Fluor 647; Molecular Probes/Thermo Fisher Scientific) and DAPI solution (Sigma-Aldrich) were used at a concentration of 1:1,000 and 600 nM, respectively, and washed three times in PBT before they were mounted in Aqua Poly/Mount (Polysciences). Immunostained embryos were imaged using a Zeiss LSM 780 confocal microscope with a Plan Apochromat 63×/NA 1.2 water immersion objective (Carl Zeiss) at 20°C using Zen Black software (Zeiss).

### Cell culture, transfection, and live imaging
HeLa cells were cultured in DMEM (Gibco) supplemented with 2 mM L-glutamine (Gibco) and 10% fetal calf serum (PAA) at 37°C and 5% CO$_2$. The cells were grown on 35-mm tissue culture–ready glass-bottom dishes (MatTek). Plasmid transfection was done using Fugene HD (Roche) according to the manufacturer's instructions. Cells were live-imaged using a Zeiss LSM 780 confocal microscope with a Plan Apochromat 63×/NA 1.2 water immersion objective (Carl Zeiss) at 37°C and 5% CO$_2$ using Zen Black software (Zeiss).

### Cloning
All plasmids were cloned using Gibson assembly (Gibson et al., 2009), the Gateway cloning system, and standard molecular biology procedures. *bnk* (UniProtKB accession number P40794; European Nucleotide Archive reference sequence AAC46467.1) coding sequence was amplified from a *Drosophila* embryo (0–12 h) cDNA library. *bnk* truncation constructs (1–303, 1–100, 100–200, 1–197, 140–251, 198–303, and 247–303) were cloned into pENTR/D-TOPO (Thermo Fisher Scientific) and then subcloned into a mammalian expression vector comprising a C-terminal GFP tag (pcDNA6.2/C-EmGFP-Dest; Thermo Fisher Scientific) using the Gateway system. A custom-made destination vector (pDEST-MBP-3C-His) containing a T7 promoter, the MBP gene, an HRV 3C site cleavage site, a 6xHis-tag, and chloramphenicol resistance together with a ccdB death cassette flanked by attachment R sites was derived

from a pDEST 17 (Thermo Fisher Scientific) vector. *bnk* fragments (1–303, 1–150, 50–197, 1–197, 50–197, 140–303, 198–303, 247–303, and 50–303) were subcloned into pDEST-MBP-3C-His to produce MBP-Bnk truncations. To generate transgenic fly lines maternally expressing the nanobody, the gene for the F-box protein Slimb fused to a nanobody gene was amplified from a pUAST-attB-nanobody vector (gift from E. Caussinus) and cloned into a modified pCatub67Mat-polyA *Drosophila* expression vector.

## Verification of YFP-Cheerio and YFP-Fimbrin insertion sites

To verify the position of the YFP in the gene locus of *Cheerio* and *Fimbrin*, cDNA was generated from adult flies of the cher [CPTI001399] and Fim[CPTI100066] lines using gene-specific and YFP/Venus-specific primers (for *cheerio*: gfp-fwd 5′-ACG TAAACGGCCACAAGTTC-3′, cher-rev 5′-GGCTCACCGGTGACT CCGT-3′; for *fimbrin*: fim-fwd 5′-TGTATCTGAGCATCAAGGATG G-3′, gfp-rev 5′-GCTGAACTTGTGGCCGTTTA-3′; gfp-fwd2 5′-ACATGGTCCTGCTGGAGTTC-3′, fim-rev 5′-AGTTGCAGTTCT CCAGCTTTTC-3′). The insertion and surrounding genomic regions were cloned into a pCR-Blunt II-TOPO vector using the Zero Blunt TOPO PCR Cloning Kit (Thermo Fisher Scientific) and sequenced. YFP insertion in the *Fimbrin* locus was determined to be in between exons 2 and 3 after the amino acid residue L-395 of Fimbrin(RA) protein sequence. The YFP insertion in the *Cheerio* locus at 3R:17,096,706 was confirmed.

## Single embryo genotyping PCR

Fly embryos were picked and smashed in squeezing buffer (10 mM Tris-HCl, pH 7.5, 1 mM EDTA, 25 mM NaCl, and 200 µg/ml Proteinase K; Thermo Fisher Scientific) and incubated for 30 min at 37°C. Primer pairs specific for *bnk* (5′-ACTGGGATCCAT GAGCATCAGCACTTTCAACTTCC-3′ and 5′-AGTCGCGGCCGCCTA AGCACTCATCGAGATGCGCTGC-3′) and *cheerio::YFP* (5′-CCAGGA GCGCACCATCTTCTTCAAGG-3′ and 5′-GCCCTTGTACAGCTCGTC CATGCCG-3′) were used to amplify gene-specific fragments by PCR.

## Fly strains and genetics

All fly stocks were kept at 22°C or 18°C.

The following endogenously YFP-tagged fly lines were generated by the Cambridge Protein Trap Insertion (CPTI) project and obtained from the Kyoto *Drosophila* Genomics and Genetic Resources stock center: for Cheerio::YFP: w[1118];+; cher[w+, CPTI001399] (stock: 115514); for Fimbrin::YFP: w[1118], Fim[w+, CPTI100066] (stock: 115092).

A fly line carrying a deficiency deleting the *bnk* gene was obtained from the Bloomington stock center: w1118; Df(3R) Exel6218/TM6B,Tb (stock: 7696).

To visualize myosin-II, the following steps were taken: yw[*]; P[w+, sqhp>sqh::mCherry]/CyO; Dr/TM3, Ser, Sb]. Myosin regulatory light chain (spaghetti squash) was tagged with the fluorescent protein mCherry.

Two transgenic lines expressing the anti GFP-nanobody under the maternal tubulin promoter and spaghetti squash (*sqh*) promoter were generated by P-element transformation and balanced to obtain two genotypes: w[*]; P[w+, mat.tub> Slimb-GFP-nanobody]/CyO and w[*]; P[w+, sqh> Slimb-GFP-nanobody]/CyO.

When expressed in flies homozygous for Cheerio::YFP, both anti GFP-nanobody–expressing fly lines resulted in protein depletion in early embryos. However, the fly line expressing anti GFP-nanobody under the *sqh* promoter resulted in sterility, likely due to defects during oogenesis. For this reason, only w[*]; P[w+, mat.tub> Slimb-GFP-nanobody]/ CyO was used in this study.

A transgenic line expressing the anti GFP-nanobody under the maternal tubulin promoter was generated by microinjection and standard procedures: w[*]; P[w+, mat.tub> Slimb-GFP-nanobody]/CyO.

To visualize myosin-II upon protein KD of Cheerio, the following line was generated: w[*]; P[w+, mat.tub> Slimb-GFP-nanobody]/P[w+, sqhp>sqh::mCherry]; cher[w+, CPTI001399].

To visualize myosin-II upon protein KD of Fimbrin, the following line was generated: w[1118], Fim[w+, CPTI100066]; P[w+, mat.tub> Slimb-GFP-nanobody]/P[w+, sqhp>sqh:: mCherry];+.

To visualize Cheerio::YFP and the nuclei in a *bnk*$^{-/-}$ embryo: w[*]; P[w+, His2Av-mRFP]; cher[w+, CPTI001399]/Df(3R) Exel6218.

To visualize myosin-II upon protein KD of Cheerio in combination with *bnk*$^{-/-}$, cher[w+, CPTI001399] and Df(3R) Exel6218 were recombined to generate the following lines: w[*]; P[w+, mat.tub> Slimb-GFP-nanobody]/P[w+, sqhp>sqh:: mCherry]; cher[w+, CPTI001399], Df(3R)Exel6218; cher[w+, CPTI001399].

To visualize myosin-II upon protein KD of Fimbrin in combination with the *bnk*, deficiency the following lines were generated: w[1118], Fim[w+, CPTI100066]; P[w+, mat.tub> Slimb-GFP-nanobody]/P[w+, sqhp>sqh::mCherry]; Df(3R)Exel6218/TM3, P{GAL4-twi.G}2.3, P{UAS-2xEGFP}AH2.3, Sb, Ser (the third chromosome balancer contains a transgene expressing GAL4 under the *twist* promoter and EGFP under the UAS promoter, Bloomington stock center; stock: 51328).

For optogenetic experiments and visualization of the myosin-II probe Sqh::mCherry, the following females were generated and crossed to wild-type males and maintained in the dark: w[*]; P[w+, UASp>CIBN::pmGFP]/P[w+, Sqh::mCherry]; P[w+, UASp>Rho-GEF2-CRY2]/P[w+, Osk>Gal4::VP16].

For optogenetic experiments in *bnk*$^{-/-}$ mutant embryos, the myosin-II probe Sqh::mCherry and Osk>Gal4 on the second chromosome were recombined to obtained the genotype w[*]; P[w+, UASp>CIBN::pmGFP]/P[w+, Sqh::mCherry],P[w+, Osk>Gal4::VP16]; P[w+, UASp>RhoGEF2-CRY2]/Df(3R)Exel6218.

For the optogenetic experiments, the following stocks were used: w[*]; If/CyO; P[w+, UASp>RhoGEF2-CRY2]/TM3, Ser. Gal4-UAS-driven *Drosophila* RhoGEF2 DHPH domain fused to CRY2 PHR domain, and w[*]; P[w+, UASp>CIBN:: pmGFP]/Cyo; Sb/TM3, Ser. Gal4-UAS-driven CIBN fused to EGFP and CAAX box (membrane anchor) and w[*]; If/CyO; P[w+, Oskp>Gal4::VP16]/TM3, Ser and w[*];P[w+, Oskp>Gal4:: VP16]/CyO;Sb/TM3, Ser. *Oskar* promoter-driven and maternally deposited Gal4 transcription factor (Bloomington stock center; stock: 44242).

## Online supplemental material

Fig. S1 shows the differential behavior of the basal actomyosin network during different stages of cellularization upon optogenetic stimulation of myosin-II in control and bnk$^{-/-}$ embryos. Fig. S2 shows an in silico analysis of Bnk and actin cosedimentation assays using α-actinin and BSA controls. Fig. S3 schematically depicts the position of YFP insertions in cheerio and fimbrin and data analyzing Cher and Fim protein localization and KD efficiency during cellularization. Fig. S4 shows an analysis of actin fiber width and density of STED images. Fig. S5 shows verifications of the bnk mutant genotype in Cheerio and Fimbrin KD embryos. Video 1 shows laser ablation experiments during the hexagonal and ring phase in nonactivated control embryos and upon photoactivation. Videos 2, 3, 4, and 5 show the basal actomyosin network during cellularization in a control embryo (Video 2), in a bnk$^{-/-}$ embryo (Video 3), in a Fimbrin KD embryo (Video 4), and in a Cheerio KD (Video 5) embryo. Video 6 shows an analysis of actomyosin network angularity in control, bnk$^{-/-}$, Fimbrin KD, and Cheerio KD embryos. Video 7 shows a control and a bnk$^{-/-}$ Cheerio KD double mutant embryo during cellularization. Video 8 shows a control and a bnk$^{-/-}$ Fimbrin KD double mutant embryo during cellularization. Video 9 shows cross sections of a control, a bnk$^{-/-}$, and a bnk$^{-/-}$ Fimbrin KD double mutant embryo during gastrulation.

## Acknowledgments

We thank all members of the De Renzis laboratory for helpful discussion. We thank J. Crocker and A. Ephrussi for critical reading of the manuscript and L. Blanchoin for advice on in vitro actin binding/bundling assay. We thank E. Loeser (European Molecular Biology Laboratory, Heidelberg, Germany) for generating the maternal tubulin nanobody fly line and E. Caussinus (University of Basel, Basel, Switzerland) for providing the nanobody plasmid. We thank the advanced light microscopy core facility (European Molecular Biology Laboratory) and the protein expression and purification facility (European Molecular Biology Laboratory) for their advice and assistance. We thank the Bloomington Drosophila Stock Center for providing fly stocks and the Drosophila Genomics Resource Center for providing cDNAs.

This work was supported by European Molecular Biology Laboratory core funding.

The authors declare no competing financial interests.

Author contributions: The experiments were conceived and designed by D. Krueger and S. De Renzis. All experiments were performed by D. Krueger except the EM experiments described in Fig. 2, which performed by S.A. Mortensen and analyzed by S.A. Mortensen and C. Sachse. T. Quinkler performed the actin binding/bundling assays described in Fig. 2 and together with D. Krueger developed the strategy to express and purify Bnk. All the data were analyzed by D. Krueger and S. De Renzis. D. Krueger wrote the manuscript with input from S. De Renzis, C. Sachse, and S.A. Mortensen.

Submitted: 22 November 2018

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
