## [Reviewer comments · The Journal of Cell Biology]

Crosslinker-mediated regulation of actin network organization controls tissue morphogenesis

Daniel Krueger, Theresa Quinkler, Simon Arnold Mortensen, Carsten Sachse, and Stefano De Renzis

Corresponding Author(s): Stefano De Renzis, European Molecular Biology Laboratory

Review Timeline:	Submission Date:	2018-11-22
	Editorial Decision:	2019-01-04
	Revision Received:	2019-04-29
	Editorial Decision:	2019-05-22
	Revision Received:	2019-05-29

Monitoring Editor: Ewa Paluch

Scientific Editor: Tim Spencer

Transaction Report:

DOI: <https://doi.org/N/A>

January 4, 2019

Re: JCB manuscript #201811127

Dr. Stefano De Renzi
European Molecular Biology Laboratory
Meyerhofstrasse,1
Heidelberg 69117
Germany

Dear Dr. De Renzi,

Thank you for submitting your manuscript entitled "Crosslinker regulation of actin network topology controls tissue morphogenesis". The manuscript was assessed by expert reviewers, whose comments are appended to this letter. We invite you to submit a revision if you can address the reviewers' key concerns, as outlined here.

You will see that while all three reviewers have displayed enthusiasm for the paper, they each raise a number of concerns that will need to be addressed before the paper can be accepted for publication. We hope that you will be able to address all of the main reviewer comments with new experiments and/or modifications to the text. Please note, however, that while we agree with reviewer #2 that assessing phase-specific Rho activation state would be an interesting avenue of exploration, we do not feel that these experiments are explicitly necessary to support the main conclusions so we will leave the decision to you whether you address this point experimentally or via added discussion.

GENERAL GUIDELINES:

Text limits: Character count for an Article is < 40,000, not including spaces. Count includes title page, abstract, introduction, results, discussion, acknowledgments, and figure legends. Count does not include materials and methods, references, tables, or supplemental legends.

Figures: Articles may have up to 10 main text figures. Figures must be prepared according to the policies outlined in our Instructions to Authors, under Data Presentation, <http://jcb.rupress.org/site/misc/ifora.xhtml>. All figures in accepted manuscripts will be screened prior to publication.

IMPORTANT: It is JCB policy that if requested, original data images must be made available. Failure to provide original images upon request will result in unavoidable delays in publication. Please ensure that you have access to all original microscopy and blot data images before submitting your revision.

Supplemental information: There are strict limits on the allowable amount of supplemental data.

Articles may have up to 5 supplemental figures. Up to 10 supplemental videos or flash animations are allowed. A summary of all supplemental material should appear at the end of the Materials and methods section.

The typical timeframe for revisions is three months; if submitted within this timeframe, novelty will not be reassessed at the final decision. Please note that papers are generally considered through only one revision cycle, so any revised manuscript will likely be either accepted or rejected.

Thank you for this interesting contribution to Journal of Cell Biology. You can contact us at the journal office with any questions, cellbio@rockefeller.edu or call (212) 327-8588.

Sincerely,

Ewa Paluch, PhD
Monitoring Editor
JCB

Tim Spencer, PhD
Deputy Editor
Journal of Cell Biology
ORCID: 0000-0003-0716-9936

Reviewer #1 (Comments to the Authors (Required)):

This study demonstrates the importance of the regulation of actin bundling activity in order to tune myosin-induced contractility, required to achieve proper cellularization of the *Drosophila* early embryo. Using both a detailed in vivo quantification together with an optogenetic induction of contractility or the study of mutant phenotypes, as well as in vitro biochemical characterization, the authors show that the presence of multiple molecular players enables a fine spatio-temporal control of actin architecture and contractility, thus closely dictating cell shape changes. This study confirms in a live system previous results obtained from theoretical and in vitro studies. It underlies the importance of a proper regulation of actin dynamics and architectures during a specific morphogenesis event.

First the authors adapted light controlled Rho1 activation at the membrane to increase locally contractility in a spatio-temporal controlled manner. This experiment is key to the study as it enables to conclude that there is temporal control of actin network's ability to contract.

The authors then directly show the biochemical characterization of Bottleneck protein. Bnk acts as an actin bundling protein. At this stage I find however the conclusion p9 (Bnk bundling, "likely antagonizes myosin" induced contractility) too overstating. The mutant phenotype for instance

supporting this data is shown later.

Using deGradFP protein knockdown the authors observe the impact on cell shape dynamics of the independent reduction of three proteins regulating actin bundling. Bnk and Cheerio(Filamin) having a similar impact of increasing cell rounding and contraction while Fimbrin(Plastin) having an opposite effect, enhancing hexagonal state. Using super resolution microscopy (STED) the authors study actin organization in the hexagonal and ring phases in WT and protein depleted conditions. If the images shown in Figure 4 shows significative differences, proper quantification is here missing and the corresponding description is not convincing in the results section. For instance, the sentence "as an array of individual filaments which are tightly bundled together and interconnected across the entire basal surface" p11 is to my opinion unclear and not right. Density, width, straightness of bundles and actin architectures can easily be further quantified here to back up conclusions. I am also a bit puzzled to have so much mention of connectivity while commenting static images. Could the author image Bnk, Cheerio and Fimbrin in STED too?

Important too to consider: the orientation of the observation plane between hexagonal and ring is different (hexagonal is a cross section of the actin architecture and rings are in the plane... this would need to be taken into account while describing the architecture.

Finally, the authors study the cooperative role between Cheerio and Bnk. Data indicates rather parallel pathways. Only ingression depth is analyzed in this section, what about angularity or area? By eye no size difference is observed between F and I.

What about the phenotype of a simultaneous KD of Cheerio and Fimbrin ?

p12 first sentence: I am not sure I have at this point the data showing the "Bnk antagonizes Fimbrin activity in promoting ring formation". What about Cheerio too ?

Minor Comments:

Abstract, the introduction of bottleneck is a bit abrupt, add some info on Bottleneck such as no equivalent in other organism etc.

Figure 1: axis unclear in the M and O panel, in addition to fold change, should be written for example area and intensity level for a quicker understanding

Figure 2: in the text add error bar for average width (p9 ex 10nm +/- x nm) data from Qpanel

Figure 5-6, some information on z are missing

Reviewer #2 (Comments to the Authors (Required)):

In this study, Krueger and colleagues study the effect of actin organization on early drosophila embryo cellularization, and its impact on the contraction of the basal actomyosin meshwork. The slow phase of cellularization is associated with actomyosin organization in hexagons that are poorly contractile, while the fast phase of cellularization is associated with a reorganization in rings and faster constriction. Using a recent optogenetic tool developed by their lab (membrane relocalisation of the GEF RhoGEF2) they could show that the early meshwork is much less responsive to MyoII recruitment (little contraction) compared to the late network. They then try to explain this difference in behavior. Previously, the gene Bottleneck (identified by E Wieshauss lab in

1993) was shown to regulate this transition from hexagonal to circular array. More specifically, Bottleneck mutants show premature constriction and premature reorganization of the cytoskeleton, and Bottleneck protein disappears from the furrow at the onset of the fast phase (Schejter and Wieshauss 1993). Previous observations already suggested that Bottleneck interacts with actin and that it could help to reorganize the cytoskeleton, however there was no biochemical characterization of this interaction. Using purified proteins and purified actin, the author clearly demonstrate in this study that Bottleneck protein is sufficient to bundle actin filaments in vitro. They then screen for actin regulators localized at the furrow and found that Cheerio (a filamin) and Fimbrin both localize to the basal actomyosin network. Using tools to downregulate Cheerio and Fimbrin proteins, they show very convincing experiments that establish the epistatic relationship between Bottleneck, Fimbrin and Cheerio : while Cheerio and Bottleneck are required to prevent early constriction, Fimbrin promotes the transition to a contractile network. The most striking result is the rescue of the actomyosin organization back to WT in the Bottleneck-Fimbrin double knockdown.

In the last few years, most of the studies related to morphogenesis in vivo have been focusing on the recruitment, the polarity and the dynamics of the molecular motor Myosin II; mostly regulated biochemically by kinases and phosphatases. Yet, in the past years there has been a growing body of evidences showing both in vitro and in cell culture that actin organization/dynamics (crosslinking, filament length, and turnover) can also affect tension independently of changes in MyoII activation/concentration. In that regard, this study provides very interesting validation of this concept in vivo. Moreover, it brings a new piece to the old unresolved question of how the transition of the acto-myosin mesh during cellularisation is controlled (although there was previous studies characterizing other actin regulators involved in this transition) and how Bottleneck regulates this transition (more than 20 years after its discovery). I therefore believe that this manuscript could be of very broad interest for the cell biology and the developmental biology community. Moreover, the experiments and results are overall very convincing.

I have though a several concerns that would need to be addressed. Most of them are minors and would need clarification, some would require additional (but hopefully reasonable) experiments.

Main concern:

The key experiment of this manuscript (shown in Fig. 1) is the low level of contraction of the hexagonal array despite MyoII recruitment. This is especially relevant if one want to argue that actin organization rather than MyoII levels regulates contractility. I therefore believe that this point would need to be reinforced.

While the experiment is by itself convincing, there could different way to interpret this low contractility, and this was not really discussed in the manuscript. So far, I can see 3 non-exclusive explanations:

1. The low contractility at earlier stage may be associated with lower activation of Rho. For instance, one could imagine a stage specific recruitment of some RhoGAP which could prevent strong Rho activation at early phase. Could the author use some Rho sensor to compare the levels of activation at early an late phase after RhoGEF2 activation ? The similar levels of recruitment of MyoII at both stages probably argue against this point, but it would for sure deserve some discussion.

2. The tension of the mesh is the same at early and late phase, but some factors external to the actomyosin meshwork resist to the deformation. For instance, nuclei mechanical properties could change over the course of cellularization and may be more or less permissive to contraction or the deformability may be different in the center of the nucleus (hexagonal phase) compared to its

extremities (ring phase). Here again, the fact that Bottleneck mutant can strongly deform nuclei argue against this, but this would deserve more discussion.

3. Finally, the low contraction is due to the intrinsic organization of the actomyosin network (as suggested by the author)

To complement those points, could the authors perform laser ablation of the mesh to evaluate its tension after MyoII recruitment induced by the opto Rhogef 2 ? One may expect different tensions in the early and later mesh despite similar levels of MyoII.

Also, could the author discuss the potential function of Myosin II phosphatase in this differential response ? If I am correct, previous reports have shown a progressive decrease of MyoII phosphatase levels in between those two phases (Xue and Sokac 2016)

Minor points:

- In the introduction, the authors seem to suggest that the link between actin organization and tension was only analysed in vitro. However there was very convincing evidences from Ewa Paluch lab correlating actin filament length and cortical tension in cell culture (Chug P et al, NCB 2017). This is not related to crosslinker, but I believe this work should be quoted somewhere in the introduction.

- Several quantifications were not very clearly explained in the methods. For instance, it was not clear to me how the area of the contractile ring was quantified, especially at the late phase or in Bottleneck mutants where MyoII signals seems pretty homogenous and dispersed all around the nuclei (e.g.: figure 3P). Did the authors use the bright pixels to segment the ring or (as I would rather do based on the picture) did they use inverted intensity to segment the black holes left by the nuclei ? Similarly, the way MyoII intensity fold change is measured is not clear (what is the time window used ? Do you first segment to exclude the nuclear zone ?). Those quantifications are at the basis of key results of the paper and would deserve more explanations.

- Fig. 1N : I guess the unit should be $\mu\text{m}^2/\text{min}$ (and not $\mu\text{m}/\text{min}$)

- Fig. 2Q : Could you show a scatter plot instead of an histogram ? (it would be interesting to get information about the real distribution)

- Fig 4: It may help to put somewhere on the figure that this is actin/phalloidin.

- It would be interesting to get quantification of the Cheerio and Fimbrin intensity on the basal cortex over time (is it pretty homogenous or are there stage specific accumulation ?)

- Surprisingly, cellularization and mesh reorganization seems pretty normal in the Bottleneck Fimbrin double knockdown. While this is a very striking result, it also clearly suggests that components independent of Bottleneck can also regulate the mesh reorganization. Could the author discuss a bit more this point (especially in the light of all the other actin regulators having a role in this process: anilin, septins...) ?

- Have the authors tried to look at MyoII upon Fimbrin and Cheerio double depletion ? Does it also rescue Cheerio depletion phenotype ? (if the genetic is difficult with the , the author could try to deplete them with Trip RNAi)

- Could the authors test the effect of the optogenetic activation of MyoII at the early phase in the Bottleneck or the Cheerio depleted background (one would expect contraction similar/higher than activation of MyoII in WT embryo at ring phase) ?

Reviewer #3 (Comments to the Authors (Required)):

SUMMARY

The manuscript presents a strong combination of experimental approaches to explore an understudied and challenging problem, the roles of actin and actin binding proteins in cell and tissue morphogenesis in vivo. They find important functions for three actin binding proteins (Bottleneck, filamin, fimbrin) in organizing actomyosin behavior during *Drosophila* cellularization. They provide interesting optogenetics experiments to probe contractile properties of actin networks in vivo. They characterize novel actin bundling activity of bottleneck in vitro. The authors provide a potential model explaining their results, which argues for a key role for actin crosslinkers in spatiotemporally controlling actomyosin contractile behavior. The findings will be of interest to many readers of JCB who study the cytoskeleton, actomyosin contractility, and morphogenesis. However, I find that several of the main conclusions of the paper are not fully supported by the experimental results (details in major comments below).

MAJOR COMMENTS

OPTOGENETICS EXPERIMENT:

- The optogenetics experiments in Figure 1 are interesting, but it is not clear that the only interpretation of the results is that actin in the hexagonal configuration is more resistant to myosin-driven contraction. For example, couldn't mechanical resistance (e.g. from the nucleus) explain the lower constriction rate for the hexagonal phase in Figure 1N? As such, it would be ideal (if possible) to show that optogenetic activation of myosin drives faster contraction in the Bottleneck mutant and/or Cheerio knockdown.

ACTIN CROSSLINKING AND NETWORK ORGANIZATION:

- Figure 4. It is not clear that the authors can distinguish individual actin filaments here. For example, in Fig 1C it is not clear to me if the bright regions are bundles of actin filaments or instead represent regions of high concentrations of more disordered actin networks. As actin organization is a major point in the paper, this deserves more discussion and justification.

- Many crosslinkers can generate actin bundles or more isotropic cross-linked actin networks depending on crosslink concentration, actin concentration, actin filament length, and actin turnover. For the in vitro bundling assays, it would be helpful to have more discussion of these experimental details so that actin bundling by Bottleneck can be more directly compared to other crosslinkers.

- In Figure 2P, it is interesting that Bottleneck only appears down the middle of the bundle. Is this expected for a typical actin crosslinker? Does Bottleneck have a predicted actin binding domain?

- Human filamin is large and often associated with isotropic crosslinked actin networks, although it can form actin bundles at high concentrations in vitro. How similar is Cheerio to human filamin? Is it surprising that it seems to play more of a bundling function here?

- The results on ventral furrow invagination in Fig 6, do not seem sufficient to justify generalizing the results to all morphogenetic processes. As such, it would seem that a more accurate title might refer specifically to the process of cellularization instead of the more general "tissue morphogenesis"

- Figure 7, given concerns above, perhaps it should be made more clear that this is a

proposed/hypothesized model.

MINOR COMMENTS

- A large body of literature exists on actin network organization and actin binding proteins in cultured cells (e.g. on stress fibers, cytokinetic rings, dendritic actin networks). This work should be mentioned and cited in the introduction.
- It would be helpful to have a brief introduction to the different phases of cellularization in the main text. For example, it was not immediately clear how "slow phase" (Top of page 7) mapped onto priming, hexagonal, and ring phases (Fig 1 A-C). Also, it would be helpful to explain in the main text how these phases were determined.
- Top of page 8, the quoted constriction rate of $\sim 6\mu\text{m}/\text{min}$ appears to be somewhat higher than shown in figure 1N.
- Since RhoGEF2-Cry2 could diffuse after light activation, there could be increased cortical myosin accumulation outside the activation plane, which could influence cellularization. Can this be ruled out?
- In addition to controlling myosin II activity, is there a potential for the RhoGEF2-Cry2 tool to influence other aspects of actin cytoskeleton behavior?
- It would be helpful to more clearly define "contractile units" in the main text as the authors seem to be using this term differently than it used in other literature, for example in papers on stress fibers or cytokinetic rings.
- How penetrant are phenotypes for Bottleneck mutants, fimbrin knockdown, Cheerio knockdown?
- It would be helpful to the reader to have numbers of expts/embryos/cells listed in each figure caption.
- Both the SD and SEM are used in manuscript. The authors should clarify if these are calculated based on numbers of embryos or numbers of contractile units/cells.
- While the manuscript is generally well-written, a large number of spelling errors and typos should be corrected.

Rebuttal

We thank the reviewers for their suggestions and recommendations. Below written in blue colour, our point-by-point response to the comments that have been raised. In the manuscript, new text is in red.

Reviewer #1:

This study demonstrates the importance of the regulation of actin bundling activity in order to tune myosin-induced contractility, required to achieve proper cellularization of the *Drosophila* early embryo. Using both a detailed in vivo quantification together with an optogenetic induction of contractility or the study of mutant phenotypes, as well as in vitro biochemical characterization, the authors show that the presence of multiple molecular players enables a fine spatio-temporal control of actin architecture and contractility, thus closely dictating cell shape changes. This study confirms in a live system previous results obtained from theoretical and in vitro studies. It underlies the importance of a proper regulation of actin dynamics and architectures during a specific morphogenesis event.

First the authors adapted light controlled Rho1 activation at the membrane to increase locally contractility in a spatio-temporal controlled manner. This experiment is key to the study as it enables to conclude that there is temporal control of actin network's ability to contract.

The authors then directly show the biochemical characterization of Bottleneck protein. Bnk acts as an actin bundling protein. At this stage I find however the conclusion p9 (Bnk bundling, "likely antagonizes myosin" induced contractility) too overstating. The mutant phenotype for instance supporting this data is shown later.

We agree with this comment and taking into account an additional point raised by reviewer #3, this sentence has been replaced with (see p. 11): "although we did not test whether different concentrations of Bnk could induce other types of actin organization, the results presented above demonstrate that purified Bnk₁₉₈₋₃₀₃ induces actin bundling/crosslinking. Together with the premature contraction phenotype characteristic of *bnk* ^{-/-} mutants (Schejter and Wieschaus, 1993a), these results further suggest that Bnk might antagonize myosin-II mediated contractile forces during the slow phase of cellularization by inducing actin bundling/crosslinking". By adding the reference to the Bnk mutant phenotype (premature constriction) described by Schejter et al., we think it is more evident why we put forward the hypothesis that Bnk counteracts contractility by inducing actin bundling already at this stage of the manuscript.

Using deGradFP protein knockdown the authors observe the impact on cell shape dynamics of the independent reduction of three proteins regulating actin bundling. Bnk and Cheerio(Filamin) having a similar impact of increasing cell rounding and contraction while Fimbrin(Plastin) having an opposite effect, enhancing hexagonal state. Using super resolution microscopy (STED) the authors study actin organization in the hexagonal and ring phases in WT and protein depleted conditions. If the images shown in Figure 4 shows significative differences, proper

quantification is here missing and the corresponding description is not convincing in the results section. For instance, the sentence "as an array of individual filaments which are tightly bundled together and interconnected across the entire basal surface" p11 is to my opinion unclear and not right. Density, width, straightness of bundles and actin architectures can easily be further quantified here to back up conclusions. I am also a bit puzzled to have so much mention of connectivity while commenting static images. Could the author image Bnk, Cheerio and Fimbrin in STED too?

Quantification of STED data (previous Fig.4 now Fig.5) are presented in Figure S4. We have quantified the width and the intensity of actin (as revealed by phalloidin staining) during the hexagonal phase in wild type and mutant embryos. These results are described in the results (see p.13): " While during the hexagonal phase the width of actin fibers (measured by segmenting the STED images presented above) in wild type or in Fimbrin KD embryos was ~200 nm, in *bnk* *-/-* or Cheerio KD embryos this value dropped to half (Fig. S4A-E). Segmented actin fibers in wild type and Fimbrin KD embryos displayed similar actin intensity values which were significantly higher than in *bnk* *-/-* (~2-fold) or Cheerio KD embryos (~1.5 fold) (Fig. S4F)." We also quantified the straightness of the segmented actin fibers and we did not score any significantly difference among the different conditions, therefore we did not comment on this point further.

The sentence "as an array of individual filaments which are tightly bundled together and interconnected across the entire basal surface" has been more carefully rephrased as (see p.13): "In wild type embryos during the hexagonal phase, actin appeared organized as an array of bundle-like fibers tightly juxtaposed across the entire basal surface of the embryo (although at this resolution (15 nm) we could not unambiguously distinguish between bundled actin filaments or a high concentration of disordered branched networks) (Fig. 5A-C and Fig. S5A-C)."

We have also imaged Bnk, Cheerio and Fimbrin in STED (see image below). However, we would prefer not to include this figure in the manuscript as we have already added several new data - requested by the reviewers - and have reached the limit of supplementary figures.

I

important too to consider: the orientation of the observation plane between hexagonal and ring is different (hexagonal is a cross section of the actin architecture and rings are in the plane... this would need to be taken into account while describing the architecture.

The orientation of the observation plane between hexagonal and ring is not different. It's the same plane (i.e. transverse cross-section), see also cartoon in Fig. 1 A-C or attached Figure from (Pieknya A.J. et al. (2010)).

Finally, the authors study the cooperative role between Cheerio and Bnk. Data indicates rather parallel pathways. Only ingression depth is analyzed in this section, what about angularity or area? By eye no size difference is observed between F and I.

We have added new data in Fig. 6 (previous Fig. 5). In Fig. 6J we have quantified area over time and show that in double Cheerio/Bnk mutants the basal network constrict much faster than in either Cheerio KD or Bnk mutant alone. The angularity could not be measured due to the severe disorganization of the network in double mutant embryos.

What about the phenotype of a simultaneous KD of Cheerio and Fimbrin ?
p12 first sentence: I am not sure I have at this point the data showing the "Bnk antagonizes Fimbrin activity in promoting ring formation". What about Cheerio too ?

We could not analyse double Cheerio and Fimbrin KD embryos as this perturbation caused defects during oogenesis with very few embryos laid by double mutant female flies.

The sentence : "Bnk antagonizes Fimbrin activity in promoting ring formation" has been removed from p12 and moved to the end of the result section.

Minor Comments:

Abstract, the introduction of bottleneck is a bit abrupt, add some info on Bottleneck such as no equivalent in other organism etc.

We added the following sentence to the abstract: "This transition is controlled by Bottleneck, a *Drosophila* unique protein expressed for only a short time during early cellularization, which we show regulates actin bundling"

Figure 1: axis unclear in the M and O panel, in addition to fold change, should be written for example area and intensity level for a quicker understanding

Done. We have changed this to "area fold change" and "myosin-II intensity fold change" and clarified in the corresponding legend.

Figure 2: in the text add error bar for average width (p9 ex 10nm +/- x nm) data from Q panel

Done. See p. 11 : "In the absence of Bnk₁₉₈₋₃₀₃ actin appeared as single filaments with an average width of ~10 nm ± 1.6 nm (Fig. 3K, M, O, Q). The presence of Bnk₁₉₈₋₃₀₃ induced the formation of thicker actin bundles with an average width of 30 nm ± 10 nm (Fig. 3L, N, P, Q)..."

Figure 5-6, some information on z are missing

z-plane values have been added to the corresponding panels in Figure 6-7.

Reviewer #2:

In this study, Krueger and colleagues study the effect of actin organization on early drosophila embryo cellularization, and its impact on the contraction of the basal actomyosin meshwork. The slow phase of cellularization is associated with actomyosin organization in hexagons that are poorly contractile, while the fast phase of cellularization is associated with a reorganization in rings and faster constriction. Using a recent optogenetic tool developed by their lab (membrane relocation of the GEF RhoGEF2) they could show that the early meshwork is much less responsive to MyoII recruitment (little contraction) compared to the late network. They then try to explain this difference in behavior. Previously, the gene Bottleneck (identified by E Wieshauss lab in 1993) was shown to regulate this transition from hexagonal to circular array. More specifically, Bottleneck mutants show premature constriction and premature reorganization of the cytoskeleton, and Bottleneck protein

disappears from the furrow at the onset of the fast phase (Schejter and Wieshauss 1993). Previous observations already suggested that Bottleneck interacts with actin and that it could help to reorganize the cytoskeleton, however there was no biochemical characterization of this interaction. Using purified proteins and purified actin, the author clearly demonstrate in this study that Bottleneck protein is sufficient to bundle actin filaments in vitro. They then screen for actin regulators localized at the furrow and found that Cheerio (a filamin) and Fimbrin both localize to the basal actomyosin network. Using tools to downregulate Cheerio and Fimbrin proteins, they show very convincing experiments that establish the epistatic relationship between Bottleneck, Fimbrin and Cheerio : while Cheerio and Bottleneck are required to prevent early constriction, Fimbrin promotes the transition to a contractile network. The most striking result is the rescue of the actomyosin organization back to WT in the Bottleneck-Fimbrin double knockdown.

In the last few years, most of the studies related to morphogenesis in vivo have been focusing on the recruitment, the polarity and the dynamics of the molecular motor Myosin II; mostly regulated biochemically by kinases and phosphatases. Yet, in the past years there has been a growing body of evidences showing both in vitro and in cell culture that actin organization/dynamics (crosslinking, filament length, and turnover) can also affect tension independently of changes in MyoII activation/concentration. In that regard, this study provides very interesting validation of this concept in vivo. Moreover, it brings a new piece to the old unresolved question of how the transition of the acto-myosin mesh during cellularisation is controlled (although there was previous studies characterizing other actin regulators involved in this transition) and how Bottleneck regulates this transition (more than 20 years after its discovery). I therefore believe that this manuscript could be of very broad interest for the cell biology and the developmental biology community. Moreover, the experiments and results are overall very convincing.

I have though a several concerns that would need to be addressed. Most of them are minors and would need clarification, some would require additional (but hopefully reasonable) experiments.

Main concern:

The key experiment of this manuscript (shown in Fig. 1) is the low level of contraction of the hexagonal array despite MyoII recruitment. This is especially relevant if one want to argue that actin organization rather than MyoII levels regulates contractility. I therefore believe that this point would need to be reinforced. While the experiment is by itself convincing, there could different way to interpret this low contractility, and this was not really discussed in the manuscript. So far, I can see 3 non-exclusive explanations:

1. The low contractility at earlier stage may be associated with lower activation of Rho. For instance, one could imagine a stage specific recruitment of some RhoGAP which could prevent strong Rho activation at early phase. Could the author use some Rho sensor to compare the levels of activation at early an late phase after RhoGEF2 activation ? The similar levels of recruitment of MyoII at both stages probably argue against this point, but it would for sure deserve some discussion.

We discussed this possibility by adding the following sentence (p. 8): "These difference in contractility could be caused by the presence of a developmentally

controlled RhoGAP activity (Mason et al., 2016), which could prevent myosin-II activation specifically during the hexagonal phase. This is however unlikely as the levels of myosin-II were equally up-regulated in response to optogenetic activation during all phases of cellularization (Fig. 1O). It is also unlikely that myosin-II phosphatase, whose levels were previously shown to be constant until the hexagonal phase and then progressively decrease (Xue and Sokac, 2016), is responsible for these different contractile responses. Optogenetic activation during the priming phase (when myosin phosphatase is present at equal levels as during the hexagonal phase) resulted in a contractile behavior similar to the ring phase.”

2. The tension of the mesh is the same at early and late phase, but some factors external to the actomyosin meshwork resist to the deformation. For instance, nuclei mechanical properties could change over the course of cellularization and may be more or less permissive to contraction or the deformability may be different in the center of the nucleus (hexagonal phase) compared to its extremities (ring phase). Here again, the fact that Bottleneck mutant can strongly deform nuclei argue against this, but this would deserve more discussion.

We have performed optogenetic activation in *bnk*^{-/-} embryo to demonstrate that nuclei can deform even more under this condition and that lack of Bnk results in increased network responsiveness to myosin-II stimulation (Fig. S1 H-M). These results are discussed on p. 9: “Consistent with this hypothesis, optogenetic activation in a *bnk*^{-/-} embryo during early cellularization when the invaginating furrows have already passed the apices of the nuclei (corresponding to the hexagonal phase in wild type embryo), caused the network to constrict to 60% of the initial area, resembling contractile responses during the ring phase of wild type embryos (Fig. S1H-M). These results also demonstrate that lack of contractility during the hexagonal phase is not due to mechanical resistance imposed by the nuclei, which deformed in response to optogenetic activation (Fig. 1K).”

3. Finally, the low contraction is due to the intrinsic organization of the actomyosin network (as suggested by the author)

To complement those points, could the authors perform laser ablation of the mesh to evaluate its tension after MyoII recruitment induced by the opto Rhogef 2 ? One may expect different tensions in the early and later mesh despite similar levels of MyoII.

Thank you for suggesting this interesting experiment. The results are presented in an entirely new figure (Fig. 2). Briefly, we see that while there is no difference in tissue recoil upon optogenetic activation during the hexagonal phase, there is a significant increase in tissue recoil during the ring phase. This is true also in non-photoactivated embryos further supporting the hypothesis that low contraction during the hexagonal phase is due to the intrinsic organization of the actomyosin network. These results are discussed on p.9: “Furthermore, laser ablation of the basal actomyosin network during the hexagonal phase demonstrate no difference in tissue recoil between non-photoactivated and photoactivated embryos (Fig. 2A-H and Video 1), while laser ablation during the ring phase caused a 2-fold increase in tissue displacement which increased by ~16% upon optogenetic activation (Fig. 2I-N and Video 1). Taken together these results suggest that during the slow phase of cellularization, actin filaments organized in a hexagonal configuration are more resistant to myosin-II-

mediated contractile forces than when organized in different patterns during earlier and later stages of cellularization.”

Also, could the author discuss the potential function of Myosin II phosphatase in this differential response ? If I am correct, previous reports have shown a progressive decrease of MyoII phosphatase levels in between those two phases (Xue and Sokac 2016)

Done, see above point 1.

Minor points:

- In the introduction, the authors seem to suggest that the link between actin organization and tension was only analysed in vitro. However there was very convincing evidences from Ewa Paluch lab correlating actin filament length and cortical tension in cell culture (Chug P et al, NCB 2017). This is not related to crosslinker, but I believe this work should be quoted somewhere in the introduction.

This reference has been added to the introduction

- Several quantifications were not very clearly explained in the methods. For instance, it was not clear to me how the area of the contractile ring was quantified, especially at the late phase or in Bottleneck mutants where MyoII signals seems pretty homogenous and dispersed all around the nuclei (e.g.: figure 3P). Did the authors use the bright pixels to segment the ring or (as I would rather do based on the picture) did they use inverted intensity to segment the black holes left by the nuclei ? Similarly, the way MyoII intensity fold change is measured is not clear (what is the time window used ? Do you first segment to exclude the nuclear zone ?). Those quantifications are at the basis of key results of the paper and would deserve more explanations.

We have clarified these points in the relevant method section (see p. 25-26). Briefly as this referee correctly anticipated we also used inverted intensity images to segment the basal openings. Additionally, we have added a panel in Fig.1 to show the outcome of a segmentation (Fig. S1). For quantification of myosin-II intensity the time window was 4 min after photo-activation and the quantification was done on non-segmented images and background -which was estimated from the signal in the nuclear zone- was subtracted from the mean myosin-II signal present in the inter-nuclear spaces. This information has been added to the method.

- Fig. 1N : I guess the unit should be $\mu\text{m}^2/\text{min}$ (and not $\mu\text{m}/\text{min}$)

Yes, correct is $\mu\text{m}^2/\text{min}$. We have change it.

- Fig. 2Q : Could you show a scatter plot instead of an histogram ? (it would be interesting to get information about the real distribution)

Done, now Fig 3Q.

- Fig 4: It may help to put somewhere on the figure that this is actin/phalloidin.

Done. See Fig.5.

- It would be interesting to get quantification of the Cheerio and Fimbrin intensity on the basal cortex over time (is it pretty homogenous or are there stage specific accumulation ?)

These data have been added in Fig. S3F,G and discussed on p.12 : "Cheerio levels at the base increased over the course of cellularization reaching a plateau by the end of the hexagonal phase, while Fimbrin levels at the base were stable until the fast phase and decreased towards the end of cellularization (Fig. S3C-G)."

- Surprisingly, cellularization and mesh reorganization seems pretty normal in the Bottleneck Fimbrin double knockdown. While this is a very striking result, it also clearly suggests that components independent of Bottleneck can also regulate the mesh reorganization. Could the author discuss a bit more this point (especially in the light of all the other actin regulators having a role in this process: anilin, septins...)?

We have added the following sentence to the discussion (p.18): "The Fimbrin mutant phenotype described here resembles the abnormalities of anilin and septin mutant embryos (Mavrakis et al., 2014; Thomas and Wieschaus, 2004; Xue and Sokac, 2016) suggesting that a more complex molecular regulation underlies the spatio-temporal organization of actomyosin network dynamics during cellularization."

- Have the authors tried to look at MyoII upon Fimbrin and Cheerio double depletion? Does it also rescue Cheerio depletion phenotype? (if the genetic is difficult with the , the author could try to deplete them with Trip RNAi)

We could not perform this experiment as double maternal Cheerio/Fimbrin KD females did not lay eggs and we could not get consistent knockdown of Cheerio using RNAi lines.

- Could the authors test the effect of the optogenetic activation of MyoII at the early phase in the Bottleneck or the Cheerio depleted background (one would expect contraction similar/higher than activation of MyoII in WT embryo at ring phase)?

We addressed this point by performing optogenetic activation in *bnk*^{-/-} embryos. The result of this experiment is presented in Fig S1H-M and discussed in the results (see p. 9-10): "Consistent with this hypothesis, optogenetic activation in a *bnk*^{-/-} embryo during early cellularization when the invaginating furrows have already passed the apices of the nuclei (corresponding to the hexagonal phase in wild type embryo), caused the network to constrict to 60% of the initial area, resembling contractile responses during the ring phase of wild type embryos (Fig. S1H-M). These results also demonstrate that lack of contractility during the hexagonal phase is not due to mechanical resistance imposed by the nuclei, which deformed in response to optogenetic activation (Fig. 1K)."

Reviewer #3 :

SUMMARY

The manuscript presents a strong combination of experimental approaches to explore an understudied and challenging problem, the roles of actin and actin binding proteins in cell and tissue morphogenesis in vivo. They find important functions for three actin binding proteins (Bottleneck, filamin, fimbrin) in organizing actomyosin behavior during *Drosophila* cellularization. They provide interesting optogenetics experiments to probe contractile properties of actin networks in vivo. They characterize novel actin bundling activity of bottleneck in vitro. The authors provide a potential model explaining their results, which argues for a key role for actin crosslinkers in spatiotemporally controlling actomyosin contractile behavior. The findings will be of interest to many readers of JCB who study the cytoskeleton, actomyosin contractility, and morphogenesis. However, I find that several of the main conclusions of the paper are not fully supported by the experimental results (details in major comments below).

MAJOR COMMENTS

OPTOGENETICS EXPERIMENT:

- The optogenetics experiments in Figure 1 are interesting, but it is not clear that the only interpretation of the results is that actin in the hexagonal configuration is more resistant to myosin-driven contraction. For example, couldn't mechanical resistance (e.g. from the nucleus) explain the lower constriction rate for the hexagonal phase in Figure 1N? As such, it would be ideal (if possible) to show that optogenetic activation of myosin drives faster contraction in the Bottleneck mutant and/or Cheerio knockdown.

This point has been raised also by reviewer #2 and to address it we performed optogenetic activation in *bnk*^{-/-} embryos. The result of this experiment is presented in Fig S1H-M and discussed in the results (see p. 9-10): "Consistent with this hypothesis, optogenetic activation in a *bnk*^{-/-} embryo during early cellularization when the invaginating furrows have already passed the apices of the nuclei (corresponding to the hexagonal phase in wild type embryo), caused the network to constrict to 60% of the initial area, resembling contractile responses during the ring phase of wild type embryos (Fig. S1H-M). These results also demonstrate that lack of contractility during the hexagonal phase is not due to mechanical resistance imposed by the nuclei, which deformed in response to optogenetic activation (Fig. 1K)."

ACTIN CROSSLINKING AND NETWORK ORGANIZATION:

- Figure 4. It is not clear that the authors can distinguish individual actin filaments here. For example, in Fig 1C it is not clear to me if the bright regions are bundles of actin filaments or instead represent regions of high concentrations of more disordered actin networks. As actin organization is a major point in the paper, this deserves more discussion and justification.

We agree with this comment and considering also suggestions from reviewer #1 we have removed any statement referring to the visualization of individual actin filaments and we have rephrased the sentence “as an array of individual filaments which are tightly bundled together and interconnected across the entire basal surface” with (see p.13): “In wild type embryos during the hexagonal phase, actin appeared organized as an array of bundle-like fibers tightly juxtaposed across the entire basal surface of the embryo (although at this resolution (15 nm) we could not unambiguously distinguish between bundled actin filaments or a high concentration of disordered branched networks) (Fig. 5A-C and Fig. S5A-C).”

- Many crosslinkers can generate actin bundles or more isotropic cross-linked actin networks depending on crosslink concentration, actin concentration, actin filament length, and actin turnover. For the *in vitro* bundling assays, it would be helpful to have more discussion of these experimental details so that actin bundling by Bottleneck can be more directly compared to other crosslinkers.

We agree with this point and taking into account an additional point raised by reviewer #1, we added the following comment to the result (see p. 11): “although we did not test whether different concentrations of Bnk could induce other types of actin organization, the results presented above demonstrate that purified Bnk₁₉₈₋₃₀₃ induces actin bundling/crosslinking. Together with the premature contraction phenotype characteristic of *bnk* *-/-* mutants (Schejter and Wieschaus, 1993a), these results further suggest that Bnk might antagonize myosin-II mediated contractile forces during the slow phase of cellularization by inducing actin bundling/crosslinking”.

- In Figure 2P, it is interesting that Bottleneck only appears down the middle of the bundle. Is this expected for a typical actin crosslinker? Does Bottleneck have a predicted actin binding domain?

We have also wondered about this point but we have not found convincing evidence in favour or against the hypothesis that any actin crosslinker would localize specifically to the middle of the bundle as Bnk does. Therefore we decided not to speculate on this observation. Bnk does not contain any known actin binding domain.

- Human filamin is large and often associated with isotropic crosslinked actin networks, although it can form actin bundles at high concentrations *in vitro*. How similar is Cheerio to human filamin? Is it surprising that it seems to play more of a bundling function here?

To comment on this aspect we have added the following sentence to the discussion (see p. 17): “*In vitro* studies using the vertebrate homologue of Cheerio, Filamin, have demonstrated that it promotes dendritic networks at low concentrations and bundling at higher concentrations (Schmoller et al., 2009). Given the high degree of homology between Cheerio and Filamin (~50% identity with human) (Li et al., 1999; Sokol and Cooley, 1999), one possibility is that the bundling promoting activity of Cheerio relates to its relatively high concentration during cellularization.”

- The results on ventral furrow invagination in Fig 6, do not seem sufficient to justify

generalizing the results to all morphogenetic processes. As such, it would seem that a more accurate title might refer specifically to the process of cellularization instead of the more general "tissue morphogenesis"

We appreciate this concern. However, we do not think the term tissue morphogenesis in the title implies that our finding apply automatically to all morphogenetic processes. Cellularization and ventral furrow invagination are two examples of tissue morphogenesis as much as motility of macrophages is an example of cell motility. Yet, it is quite common to refer to the general term of cell motility in publication's titles rather than to the specific motility event that was studied in detail in the paper. It seems to me we all agree that one interesting aspect of our study is the link between actin crosslinkers and tissue morphogenesis, therefore we would prefer to highlight this is the title rather than referring to the specific process of cellularization. We think this will increase visibility of our study without misleading potential readers interested in the topic.

- Figure 7, given concerns above, perhaps it should be made more clear that this is a proposed/hypothesized model.

Done. Figure 7 (now Fig. 8) is presented as a cartoon and the title of its legend has now been revised to: "Model proposing how the synergistic and antagonistic role of actin crosslinkers control...". We think the term "model proposing" unambiguously states that this figure represents a working model.

MINOR COMMENTS

- A large body of literature exists on actin network organization and actin binding proteins in cultured cells (e.g. on stress fibers, cytokinetic rings, dendritic actin networks). This work should be mentioned and cited in the introduction.

We have added the following paragraph and citations to the introduction (p.3): 'In addition to myosin activation, *in vitro* contractility assays and studies in cell culture have highlighted the importance of actin crosslinkers, which connect filaments together and generate networks with different architecture and contractile properties (Blanchoin et al., 2014; Chugh et al., 2017; Koenderink and Paluch, 2018; Laporte et al., 2012; Pollard and Wu, 2010; Reymann et al., 2012; Svitkina and Borisov, 1999).

- It would be helpful to have a brief introduction to the different phases of cellularization in the main text. For example, it was not immediately clear how "slow phase" (Top of page 7) mapped onto priming, hexagonal, and ring phases (Fig 1 A-C). Also, it would be helpful to explain in the main text how these phases were determined.

We have added the following paragraph to the results (p. 7-8): "The following criteria were used to stage embryos during the different phases of cellularization. The priming and the hexagonal phases, occurring during the slow phase of cellularization, corresponded to an ingression of the invaginating furrows from 4 μm to 7 μm , respectively. During the priming phase the myosin-II signal appeared diffuse, and was not organized in a hexagonal pattern. The ring phase, which marks the beginning of the fast phase, was defined as the time point when the invaginating

furrows had passed the base of the nuclei ($>10\ \mu\text{m}$) and myosin-II had acquired a ring-like conformation”.

- Top of page 8, the quoted constriction rate of $\sim 6\ \mu\text{m}/\text{min}$ appears to be somewhat higher than shown in figure 1N.

This value refers to the median constriction rate during the ring phase and corresponds to the green box plot in Fig. 1N. We clarified the reference to the figure in the text.

- Since RhoGEF2-Cry2 could diffuse after light activation, there could be increased cortical myosin accumulation outside the activation plane, which could influence cellularization. Can this be ruled out?

All the effects we see on cellularization are compatible with basal constriction. Furthermore, when the embryos were allowed to develop further following an initial photo-activation (as for example we did in Fig. S1E) no visible defects in cellularization were observed.

- In addition to controlling myosin II activity, is there a potential for the RhoGEF2-Cry2 tool to influence other aspects of actin cytoskeleton behavior?

The only other known effect of RhoGEF optogenetic stimulation is increased actin polymerization. However, this can not explain the observed differential contractile behaviours, as actin polymerization would be equally stimulated during all phases. In addition we have now confirmed our finding also with laser cut experiments (Fig2) following recommendation of reviewer #2.

- It would be helpful to more clearly define "contractile units" in the main text as the authors seem to be using this term differently than it used in other literature, for example in papers on stress fibers or cytokinetic rings.

Thank you for this recommendation. We agree that contractile unit is an ambiguous term given its use in other context. Therefore we have removed it from the entire text and refer more specifically to basal openings and ring constrictions.

- How penetrant are phenotypes for Bottleneck mutants, fimbrin knockdown, Cheerio knockdown?

These data have been added in Fig. 4S. Briefly, Fimbrin and Cheerio knockdown phenotypes are $>70\%$ penetrant. The *bnk*^{-/-} phenotype is a complete null and is 100% penetrant (Schejter ED, Wieschaus E. Cell 1993).

- It would be helpful to the reader to have numbers of expts/embryos/cells listed in each figure caption.

Done.

- Both the SD and SEM are used in manuscript. The authors should clarify if these are calculated based on numbers of embryos or numbers of contractile units/cells.

This has been clarified in the legend.

- While the manuscript is generally well-written, a large number of spelling errors and typos should be corrected.

The entire manuscript has been now proofread by a native English speaker.

May 22, 2019

RE: JCB Manuscript #201811127R

Dr. Stefano De Renzi
European Molecular Biology Laboratory
Meyerhofstrasse,1
Heidelberg 69117
Germany

Dear Dr. De Renzi:

Thank you for submitting your revised manuscript entitled "Crosslinker regulation of actin network topology controls tissue morphogenesis". The paper has now been seen by the original reviewers and I am please to report that they all recommend acceptance of the paper (pending some further minor revisions - as you'll see below). Thus, we would be happy to publish your paper in JCB after completion of the final revisions necessary to meet both the remaining reviewer comments and our formatting guidelines (see details below).

As noted above, please be sure to address each of the remaining issues indicated by reviewers #2 and 3. Please provide a final point-by-point rebuttal to these points along with your final revision.

A. MANUSCRIPT ORGANIZATION AND FORMATTING:

Full guidelines are available on our Instructions for Authors page, <http://jcb.rupress.org/submission-guidelines#revised>. **Submission of a paper that does not conform to JCB guidelines will delay the acceptance of your manuscript.**

1) Text limits: Character count for Articles and Tools is < 40,000, not including spaces. Count includes title page, abstract, introduction, results, discussion, acknowledgments, and figure legends. Count does not include materials and methods, references, tables, or supplemental legends.

2) Figures limits: Articles and Tools may have up to 10 main text figures. You are below the limit at this time but please bear it in mind when revising.

3) Figure formatting: Scale bars must be present on all microscopy images, including inset magnifications. Molecular weight or nucleic acid size markers must be included on all gel electrophoresis - including the cropped gels in Supp figure 5A.

4) Statistical analysis: Error bars on graphic representations of numerical data must be clearly described in the figure legend. The number of independent data points (n) represented in a graph must be indicated in the legend. Statistical methods should be explained in full in the materials and methods. For figures presenting pooled data the statistical measure should be defined in the figure legends. Please also be sure to indicate the statistical tests used in each of your experiments (both

in the figure legend itself and in a separate methods section) as well as the parameters of the test (for example, if you ran a t-test, please indicate if it was one- or two-sided, etc.). Also, since you used parametric tests in your study (e.g. t-tests, ANOVA, etc.), you should have first determined whether the data was normally distributed before selecting that test. In the stats section of the methods, please indicate how you tested for normality. If you did not test for normality, you must state something to the effect that "Data distribution was assumed to be normal but this was not formally tested."

5) Title: The title should be less than 100 characters including spaces. Make the title concise but accessible to a general readership.

While your current title will be appreciated by the specialists, we do not feel that it will be accessible to a broader cell biology audience. Therefore, we suggest the following title (also incorporating the change requested by rev#3): "Crosslinker-mediated regulation of actin network organization controls tissue morphogenesis"

6) Materials and methods: Should be comprehensive and not simply reference a previous publication for details on how an experiment was performed. Please provide full descriptions (at least in brief) in the text for readers who may not have access to referenced manuscripts.

7) Please be sure to provide the sequences for all of your primers/oligos and RNAi constructs in the materials and methods. You must also indicate in the methods the source, species, and catalog numbers (where appropriate) for all of your antibodies.

8) Microscope image acquisition: The following information must be provided about the acquisition and processing of images:

- a. Make and model of microscope
- b. Type, magnification, and numerical aperture of the objective lenses
- c. Temperature
- d. imaging medium
- e. Fluorochromes
- f. Camera make and model
- g. Acquisition software
- h. Any software used for image processing subsequent to data acquisition. Please include details and types of operations involved (e.g., type of deconvolution, 3D reconstitutions, surface or volume rendering, gamma adjustments, etc.).

10) Supplemental materials: There are strict limits on the allowable amount of supplemental data. Articles/Tools may have up to 5 supplemental figures. You currently meet this requirement but please bear it in mind when revising. Please also note that tables, like figures, should be provided as individual, editable files. A summary of all supplemental material should appear at the end of the Materials and methods section.

11) eTOC summary: A ~40-50 word summary that describes the context and significance of the findings for a general readership should be included on the title page. The statement should be written in the present tense and refer to the work in the third person.

13) ORCID IDs: ORCID IDs are unique identifiers allowing researchers to create a record of their various scholarly contributions in a single place. At resubmission of your final files, please consider providing an ORCID ID for as many contributing authors as possible.

B. FINAL FILES:

-- High-resolution figure and video files: See our detailed guidelines for preparing your production-ready images, <http://jcb.rupress.org/fig-vid-guidelines>.

Thank you for this interesting contribution, we look forward to publishing your paper in Journal of Cell Biology.

Sincerely,

Ewa Paluch, PhD
Monitoring Editor
JCB

Reviewer #1 (Comments to the Authors (Required)):

As stated earlier, this study demonstrates the importance of the regulation of actin bundling activity in order to tune myosin-induced contractility, required to achieve proper cellularization of the *Drosophila* early embryo. Using both a detailed *in vivo* quantification together with an optogenetic induction of contractility or the study of mutant phenotypes, as well as *in vitro* biochemical characterization, the authors show that the presence of multiple molecular players enables a fine spatio-temporal control of actin architecture and contractility, thus closely dictating cell shape changes. This study confirms in a live system previous results obtained from theoretical and *in vitro* studies. It underlies the importance of a proper regulation of actin dynamics and architectures during a specific morphogenesis event.

To my opinion this revised manuscript answers most of the concerns raised in the previous round of revision. The authors adequately added experiments, as the combination of laser ablation and optogenetic stimulation of RhoGEF2, as well as more quantification on data sets when it was required. The authors additionally clarified several sections in the main text, such as the first result section corresponding to one of their key experiments as well as the description of the actin architecture visualized by STED microscopy. When suggested they also raised alternative explanations and discussed these options, as well as made further comparison with other systems or proteins. This allowed the authors to strengthen their conclusions, the main being the spatio-temporal organization of actomyosin networks during a key morphogenesis process modulated not by gene expression but by a developmental modulation of the actin crosslinking via Bnk.

Reviewer #2 (Comments to the Authors (Required)):

The authors have addressed convincingly all my concerns. The new data provided in this version significantly improved the manuscript. The new data on laser ablation after RhoGEF2 activation and RhoGEF2 activation in bottleneck mutants are particularly compelling and clearly demonstrate that different actin organisations are associated with different contractility. I therefore strongly recommend publication.

Very minor point :

In the figure S3F: it would help to show the line used for the kymograph in Fig S3E or by providing a little scheme explaining where the line intensity profile is taken from (it was not clear to me what was represented there).

Reviewer #3 (Comments to the Authors (Required)):

The additional experiments and analysis as well as the clarifications in the text have strengthened

the manuscript and have provided additional support to the authors' interpretation of the role of Bnk in cellularization. I do ask for the authors to 1) provide quantification of the initial retraction velocities following laser ablation in Fig 2M, as this is another standard metric for inferring tension from ablation experiments, and 2) reconsider if the term "actin network topology" in the title is appropriate given the limits of determining connectivity from these experiments - perhaps "actin network organization" is more appropriate? Otherwise, the authors have adequately addressed my concerns, and I recommend the manuscript for publication in JCB.

Rebuttal

Reviewer #1:

As stated earlier, this study demonstrates the importance of the regulation of actin bundling activity in order to tune myosin-induced contractility, required to achieve proper cellularization of the *Drosophila* early embryo. Using both a detailed *in vivo* quantification together with an optogenetic induction of contractility or the study of mutant phenotypes, as well as *in vitro* biochemical characterization, the authors show that the presence of multiple molecular players enables a fine spatio-temporal control of actin architecture and contractility, thus closely dictating cell shape changes. This study confirms in a live system previous results obtained from theoretical and *in vitro* studies. It underlies the importance of a proper regulation of actin dynamics and architectures during a specific morphogenesis event.

To my opinion this revised manuscript answers most of the concerns raised in the previous round of revision. The authors adequately added experiments, as the combination of laser ablation and optogenetic stimulation of RhoGEF2, as well as more quantification on data sets when it was required. The authors additionally clarified several sections in the main text, such as the first result section corresponding to one of their key experiments as well as the description of the actin architecture visualized by STED microscopy. When suggested they also raised alternative explanations and discussed these options, as well as made further comparison with other systems or proteins. This allowed the authors to strengthen their conclusions, the main being the spatio-temporal organization of actomyosin networks during a key morphogenesis process modulated not by gene expression but by a developmental modulation of the actin crosslinking via Bnk.

Reviewer #2 (Comments to the Authors (Required)):

The authors have addressed convincingly all my concerns. The new data provided in this version significantly improved the manuscript. The new data on laser ablation after RhoGEF2 activation and RhoGEF2 activation in bottleneck mutants are particularly compelling and clearly demonstrate that different actin organisation are associated with different contractility. I therefore strongly recommend publication.

Very minor point :

In the figure S3F: it would help to show the line used for the kymograph in Fig S3E or by providing a little scheme explaining where the line intensity profile is taken from (it was not clear to me what was represented there).

We have added boxes in Fig. S3D,E in correspondence to the area that were analysed in order to generate the kymographs in Fig. S3F

Reviewer #3 (Comments to the Authors (Required)):

The additional experiments and analysis as well as the clarifications in the text have strengthened the manuscript and have provided additional support to the authors' interpretation of the role of Bnk in cellularization. I do ask for the authors to 1) provide quantification of the initial retraction velocities following laser ablation in Fig 2M, as this is another standard metric for inferring tension from ablation experiments, and 2) reconsider if the term "actin network topology" in the title is appropriate given the limits of determining connectivity from these experiments - perhaps "actin network organization" is more appropriate? Otherwise, the authors have adequately addressed my concerns, and I recommend the manuscript for publication in JCB.

Quantification of initial velocities (between 1 and 4 sec) are not shown in Fig 2O and are in line with the displacement measurements shown in Fig. 2M,N.
We have revised the title according to this reviewer's suggestion which we also find more clear.